# Improved Representation of Agricultural Land Use and Crop Management for Large Scale Hydrological Impact Simulation in Africa using SWAT+

Albert Nkwasa[1], Celray James Chawanda[1], Jonas Jägermeyr[2,3,4], Ann van Griensven[1,5]

[1] Hydrology and Hydraulic Engineering Department, Vrije Universiteit Brussel (VUB), 1050 Brussel, Belgium
[2] NASA Goddard Institute for Space Studies, New York, NY 10025, USA
[3] Center for Climate Systems Research, Columbia University, New York, NY 10025, USA
[4] Climate Resilience, Potsdam Institute for Climate Impact Research (PIK), Member of the Leibniz Association, 14412, Potsdam, Germany
[5] Water Science & Engineering Department, IHE Delft Institute for Water Education, 2611 AX Delft, The Netherlands

*Correspondence to*: Albert Nkwasa (albert.nkwasa@vub.be)

**Abstract.** To date, most regional and global hydrological models either ignore the representation of cropland or consider crop cultivation in a simplistic way or in abstract terms without any management practices. Yet, the water balance of cultivated areas is strongly influenced by applied management practices (e.g. planting, irrigation, fertilization, harvesting). The SWAT+ model represents agricultural land by default in a generic way where the start of the cropping season is driven by accumulated heat units. However, this approach does not work for tropical and sub-tropical regions such as the sub-Saharan Africa, where crop growth dynamics are mainly controlled by rainfall rather than temperature. In this study, we present an approach on how to incorporate crop phenology using decision tables and global datasets of rainfed and irrigated croplands with the associated cropping calendar and fertilizer applications in a regional SWAT+ model for Northeast Africa. We evaluate the influence of the crop phenology representation on simulations of Leaf Area Index (LAI) and Evapotranspiration (ET) using LAI remote sensing data from Copernicus Global Land Service (CGLS) and WaPOR ET data respectively. Results show that a representation of crop phenology using global datasets leads to improved temporal patterns of LAI and ET simulations, especially for regions with a single cropping cycle. However, for regions with multiple cropping seasons, global phenology datasets need to be complemented with local data or remote sensing data to capture additional cropping seasons. In addition, the improvement of the cropping season also helps to improve soil erosion estimates, as the timing of crop cover controls erosion rates in the model. With more realistic growing seasons, soil erosion is largely reduced for most agricultural Hydrologic Response Units (HRUs) which can be considered as a move towards substantial improvements over previous estimates. We conclude that regional and global hydrological models can benefit from improved representations of crop phenology and the associated management practices. Future work regarding incorporating multiple cropping seasons in global phenology data is needed to better represent cropping cycles in areas where they occur using regional to global hydrological models.

# 1 Introduction

Even though cropland cultivation covers over 40 % of the planet's ice-free land surface, most regional and global hydrological
model applications overlook the necessity of addressing crop phenological development and/or distinguishing between
different crops (Chen and Xie, 2012; Srivastava et al., 2020). In some regional applications e.g. Schuol and Abbaspour 2006;
Schuol et al. 2008; Chawanda et al. 2020, the model applications consider one uniform generic crop as a simplification for
agricultural land use representation despite the existing wide variety of crops in agricultural land use (Sood and Smakhtin,
2015). Using a uniform generic crop for agricultural land use modelling fails to account for any variability in vegetation
attributes such as Leaf Area Index (LAI) corresponding to several crops in real world scenarios (Viña et al., 2011). Detailed
agricultural land use representation in hydrological models is important as heterogeneity in agricultural land use can have a
significant effect on hydrological fluxes such as evapotranspiration (ET) and soil moisture (Srivastava et al., 2020). Through
ET and interception (Siad et al., 2019), the water balance of agricultural land use areas is strongly influenced by the applied
management practices (e.g. planting, irrigation, fertilization, harvesting) and their precise timing (Twine et al., 2004; Raymond
et al., 2008). In the context of global change studies, realistic representation of agricultural systems is a major concern as
changes in climatic factors affect crop growth and productivity of agricultural systems (Makowski et al., 2014). Therefore,
hydrological models that simulate cropland ecosystems should have a reasonable representation of crop phenology and the
associated management practices of these ecosystems (Lokupitiya et al., 2009).

The SWAT+ model (Bieger et al., 2017; Arnold et al., 2018) which is a restructured version of SWAT (Soil and Water
Assessment Tool; Arnold et al., 1998) utilizes the principles of the EPIC crop growth model (Williams and Singh, 1995) to
simulate agricultural land by default in a generic way where the phenological development of crops from planting is driven by
accumulated heat units (Arnold et al., 1998). However, the primary controlling factor for the start of the growing season in
tropical and sub-tropical regions such as the sub-Saharan Africa is rainfall (Lotsch et al., 2003; Alemayehu et al., 2017). Waha
et al. (2013) describes the crop growing season in sub-Saharan Africa as the period in which temperature and moisture are
suitable for growth determined by the start and end of the main rainy season. Zhang et al. (2005) showed that the onset of
seasonal vegetation green-up across Africa can be directly linked to rainfall seasonality. Several Studies (e.g. Msigwa et al.,
2019, Nkwasa et al., 2020) have further pointed out how the existing multiple cropping seasons in tropical and subtropical
climates within an agricultural year coincide with the rainfall and irrigation patterns. Therefore, the use of heat units to trigger
the start of the cropping seasons could lead to inconsistencies in crop phenology simulations for tropical and sub-tropical
regions.

Croplands include various types with associated differences in crop physiology and management practices (Lokupitiya et al.,
2009; Yin and Struik, 2009). The phenological change during the vegetation cycle of crop types actively controls the ET
process through internal physiology by increasing the amount of leaf stomata with canopy growth (Gong et al., 2014). In the
SWAT+ model, plant transpiration is simulated as a linear function of Leaf Area Index (LAI) and Potential Evapotranspiration
(PET) (Neitsch et al., 2005). Thus, inconsistences in crop simulations could lead to inaccurately estimating canopy properties

such as LAI and canopy height resulting in uncertain estimates of ET (Alemayehu et al., 2016). Accurate estimations of ET in a hydrological model are important because ET is the central flux that defines land-atmosphere interactions (Mueller et al., 2011; Fisher et al., 2017).

Additionally, changes in cropland use and crop management have received little attention in hydrological impact assessments yet these may have significant impacts on model outputs (O'Neal et al., 2005). For example; Sietz et al. (2021) demonstrated the sensitivity of key hydrological processes (runoff, groundwater seepage and ET) to crop rotations in a central European region by combining a crop generator with an eco-hydrological model. By coupling a hydrological model (Variable Infiltration Capacity) with a crop growth model, soil moisture and ET were more accurately simulated by implementing crop rotations (Zhang et al., 2021). According to Abaci and Papanicolaou (2009), cropland use and cropland management practices can significantly affect the impact of rainfall on soil erosion. The crop canopy often intercepts rainfall and hinders water droplets to reduce the splash erosion through loss of speed (Hilker et al., 2014). In addition, cropland practices cause great variations in the erodibility of cropland since soil erosion depends on what crop is grown and the crop cover density (Sundborg and White, 1982). The crop cover is crucial in the estimation of the C (crop management) factor in erosion models such as the Modified Universal Soil Loss Equation (MUSLE) used by SWAT+ (Lin et al., 2014). Other crop management practices such as amounts of fertilizer, alters soil ability to produce biomass and thus alters soil resistance to erosion (Souza et al., 2017). The timing and duration of soil cover on cropland are affected by the planting and maturity dates of the crop.

Previous studies have applied the SWAT model at a regional scale within and including sub-Saharan Africa (Schuol and Abbaspour, 2006; Schuol et al., 2008). However, these studies utilized the default generic way of representing agricultural land use without any management practices. Yet, Arnold et al. (2012) emphasized the need for realistic representation of local and regional crop processes to reliably simulate the water balance, erosion and nutrient yields in a SWAT model. Chawanda et al. (2020) describes one of the few regional applications of the latest SWAT+ version in a tropical region. The study highlighted that the inclusion of irrigation and reservoirs in model set up using decision tables (Arnold et al., 2018) led to an improvement on the simulations of discharge and ET. Hence, there is need to have a proper representation of land use and agricultural processes in Africa as very few studies report on crop phenology and land use representation in SWAT (Griensven et al., 2012).

Regional cropping phenology datasets and management practices have been developed using remote sensing approaches (Li et al., 2014; Estel et al., 2016; Xiong et al., 2017) and non-remote sensing approaches, including observational census data (Potter et al., 2010; Portmann et al., 2010; Lu and Tian, 2017; Iizumi et al., 2019; Hurtt et al., 2020; Jägermeyr et al., 2021), to integrate into regional agricultural and hydrologic modelling frameworks. However, remote sensing approaches have been criticized as not being able to detect crop types and cropping sequences without local knowledge or ground truth data (Bégué et al., 2018). Nevertheless, these spatially explicit global cropping phenology data sets have not been utilized in regional hydrological models to improve the land use and crop representation.

The novelty of this study is in improving land use and crop process representation for large scale hydrological modelling using SWAT+ by (1) proposing an approach that reasonably incorporates crop phenology using decision tables and global datasets

100    of rainfed and irrigated croplands with the associated management practices in a regional SWAT+ model for Northeast Africa, (2) evaluating model improvements of crop representation by using the remote sensing LAI from Copernicus Global Land Service (CGLS) and ET derived from WaPOR (Water Productivity through Open access of Remotely sensed derived data, FAO, 2018), (3) evaluating how the consideration of crop phenology and the associated management practices affects long term water-driven soil erosion estimates. We do not intend to fully model soil erosion but show how improvements in crop

105    representation can impact soil erosion estimates.

## 2 Material and Methods

### 2.1 Study area

Our study focused on the North-eastern part of Africa (Figure 1) that covers 4,489,000 km$^2$ for a period of 7 years (2009 – 2015). This area covers wholly or partially countries of the Nile basin including Uganda, Kenya, Tanzania, Rwanda, Burundi,

110    Sudan, South Sudan, Ethiopia, Egypt.

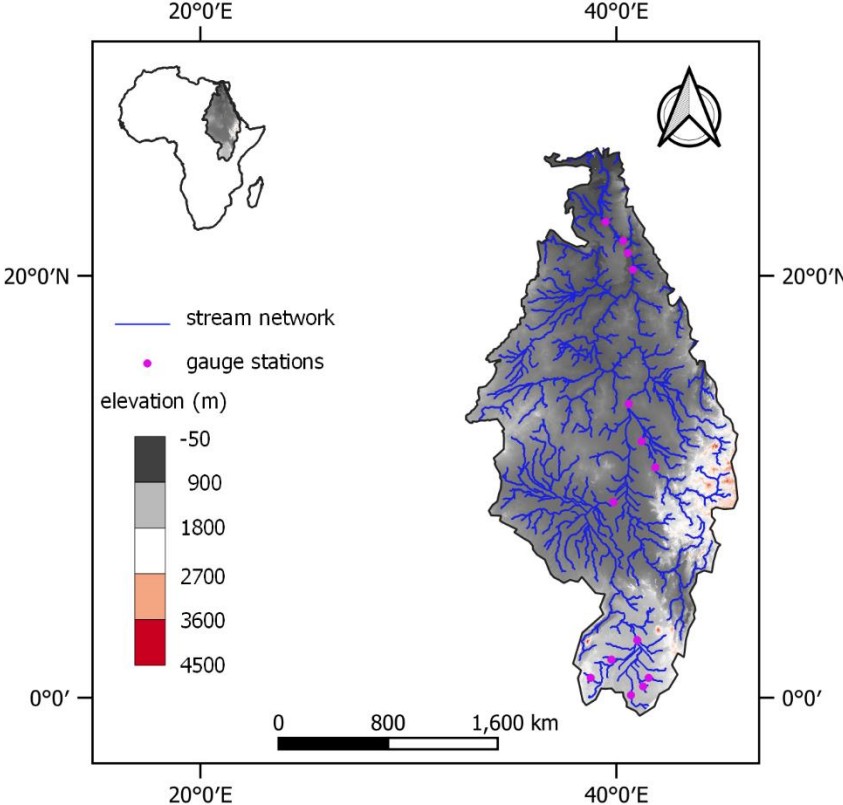

**Figure 1:** Study area - Northeast Africa (Nile basin)

The area includes the main Nile River basin with sub basins such as, Victoria Nile, Blue Nile, White Nile, Atbara, Baro-Akobo-Sobat, Bahr El jebel and Bahr El Ghazal. The major basins are the Blue Nile basin and the White Nile basin. The Blue Nile basin which originates from Lake Tana in Ethiopia is considered as the major tributary to the main Nile River due to its high contribution towards the total Nile River discharge. The White Nile starts from the great lakes region through Lake Victoria, Uganda and southern Sudan (Eldardiry and Hossain, 2019). A strong latitudinal wetness gradient characterizes the climate of the region. The areas north of 18ºN remain dry mostly of the year while there is a gradual increase of monsoon rainfall amounts in the south (Camberlin, 2009). The distribution of the mean annual rainfall is spatially contrasted in the region with about 28 % of the region receiving less than 100 mm annually. Rainfall in excess of 1000 mm y$^{-1}$ is restricted mainly to the equatorial region and the Ethiopian highlands with negligible rainfall (below 50 mm y$^{-1}$) from northern Sudan all across Egypt (Onyutha and Willems, 2015). The agricultural sector is responsible for nearly 75 % of the water withdraw within the basins (Swain, 2011). Agriculture is practiced in all elevation categories but predominantly in the low lying areas (less than 500 m) and medium areas (890 – 1450 m). Shrubland is dominant in elevation areas ranging between -47 to 1450 m and in steep slopes while forest is dominant in areas with an elevation range of 500 to 2150 m. In low lying areas, mainly in the desert area of the Nile, bare land is dominant (Nile Basin Initiative, 2016).

## 2.2 Modelling approach using SWAT+

SWAT+ is a revised version of SWAT that offers greater flexibility in connecting spatial units in the representation of management operations (Bieger et al., 2017; Arnold et al., 2018). This is a semi-distributed river basin scale model that relies on the physical characteristics of a catchment. It divides a basin into sub basins connected by a stream network, which are further divided into Hydrologic Response Units (HRUs). HRUs represent areas within the sub basin that comprise of the same land use, soil, slope and management practices (Neitsch et al., 2005). SWAT+ also introduces landscape units (LSU) to allow separation of lowland (wetland) processes from upland process (Bieger et al., 2017). SWAT+ applies the hydrological water balance concept, Eq. (1) as the basic driver of all hydrological processes.

$$WB_f = WB_i + \sum(P_j - R_j - E_j - D_j - RF_j) * \Delta t \tag{1}$$

Where; $WB_f$ and $WB_i$ are the final and initial soil water content respectively (mm d$^{-1}$), $P_j$ is the amount of rainfall (mm d$^{-1}$), $R_j$ is the amount of surface runoff (mm d$^{-1}$), $E_j$ is the ET amount (mm d$^{-1}$), $D_j$ is the percolation amount (mm d$^{-1}$), $RF_j$ is the return flow amount (mm d$^{-1}$), $\Delta t$ is the change in time (day) and j is the index. The model estimates erosion and sediment yield for each HRU using the Modified Universal Soil Loss Equation (MUSLE) (Williams and Berndt, 1977), Eq. (2). The MUSLE uses runoff energy rather than rainfall to estimate sediment yields, making it suitable at daily time scale.

$$Sed = 11.8 \, (Q_{surf} q_{peak} Area_{hru})^{0.56} \, x \, K_{USLE} \, x \, C_{USLE} \, x \, P_{USLE} \, x \, LS_{USLE} \, x \, CFRG \tag{2}$$

where; Sed is the sediment yield (tonnes/day), $Q_{surf}$ is the surface runoff volume (mm/day), $q_{peak}$ is the peak runoff rate (m$^3$/s), $Area_{hru}$ is the area of the HRU (ha), $K_{USLE}$ is the USLE soil erodibility factor, $C_{USLE}$ is the USLE crop management

factor, $P_{USLE}$ is the USLE support practice factor, $LS_{USLE}$ is the USLE topographic factor and CFRG is the coarse fragment factor.

### 2.2.1 Decision tables in SWAT+

Land use and management operations in SWAT+ can be scheduled using either or both decision tables and management schedules. However, decision tables enable the user to model intricate sets of rules and their subsequent actions by allowing them to add conditions for scheduling management (Arnold et al., 2018). Metzner and Barnes (2014) describe decision tables as a way of organizing and documenting complex events in a logical way that is easy to interpret. Nkwasa et al. (2020) compared the use of decision tables to management schedules and concluded that decision tables provided higher flexibility in representing agricultural practices. The use of decision tables in the SWAT+ model is broadly described in (Arnold et al., 2018). Scheduling in this study was done using decision tables as discussed in the subsequent sections.

### 2.2.2 Crop growth cycle with heat unit scheduling

SWAT+ uses the simplified version of the EPIC growth model to simulate plant growth (Neitsch et al., 2005). As in the EPIC model, phenological plant development is based on the daily accumulated heat units or by calendar dates, while plant growth can be inhibited by temperature, water, nitrogen and phosphorus nutrients (Neitsch et al., 2005; Arnold et al., 2012). The heat unit theory assumes that plants have requirements that can be quantified and linked to maturity. The total number of heat units required by the plant to start growing or to reach maturity is calculated as in Eq. (2).

$$PHU = \sum_{d=1}^{n}(T_{av} - T_{base}) \text{ when } T_{av} > T_{base} \tag{2}$$

where; PHU is the total heat units required to plant maturity, $T_{av}$ is the mean daily temperature ($^o$C), $T_{base}$ is the plant's minimum temperature for growth ($^o$C), $d = 1$ is the day of planting and $n$ is the number of days required for a plant to reach maturity. Planting is scheduled by a second heat index where heat units are summed over the entire year using $T_{base} = 0^oC$. This heat index is solely a function of climate calculated by SWAT+ using the provided long-term weather data (Neitsch et al., 2005). While scheduling by heat units is convenient for temperate regions that are mainly driven by temperature, users need to consider that cropping seasons in tropical and sub-tropical regions are primarily driven by water availability (Alemayehu et al., 2017). Hence, the use of heat units easily leads to incorrect cropping seasons for these regions.

### 2.3 Global datasets used for SWAT+ modelling

Modelling was done using freely available global datasets in Table 1. Of specific interest in this study is the GCCMI dataset (Jägermeyr et al., 2021) which provides a cropping calendar that is an observation-based product, combining first-hand data sources from various agricultural ministries. In the GCCMI crop calendar, planting dates and cultivar selection is based on real world observational planting and harvest data. Planting thus happens at the prescribed day per crop in each 0.5$^o$ grid cell and on average, cultivars are selected to match the observational harvest day. The development of this dataset is explicitly discussed

in (Jägermeyr et al., 2021). The climate data (EWEMBI; Lange, 2016) includes records of rainfall, maximum and minimum temperature, wind speed, solar radiation and relative humidity available at a spatial resolution of 0.5º. Irrigation data (FAO; Siebert et al., 2013) was provided as area fraction equipped for irrigation and area fraction that is actually irrigated per year. Nitrogen Fertilizer and Phosphorus fertilizer was provided as a global time series of grided synthetic fertilizer use rate in agricultural lands at a spatial resolution of 0.5º.

**Table 1:** Global datasets used for model setup

| Global Datasets | Resolution | Source |
|---|---|---|
| Digital Elevation Model (DEM) | 90 m resampled to 250 m | Shutter Radar Topography Mission (SRTM; Farr et al., 2007) |
| Land use | 0.25º | Harmonized land use (LUH2; Hurtt et al., 2020) |
| Soil | 250 m | Africa Soil information Service (AFSIS; Hengl et al., 2015) |
| Climate | 0.5º | EartH2Obseve, WFDEI and ERA-Interim data Merged and Bias corrected for ISIMIP (EWEMBI; Lange, 2016) |
| Irrigated areas | 0.083º | Food and Agriculture Organization (FAO; Siebert et al., 2013) |
| Plant and harvest dates | 0.5º | Global Gridded Crop Model Intercomparison (GGCMI; Jägermeyr et al., 2021) |
| Fertilizer – Nitrogen(N) | 0.5º | (Hurtt et al., 2020) |
| Fertilizer – Phosphorus(P) | 0.5º | (Lu and Tian, 2017) |

## 2.4 Spatial temporal analysis of rainfall

Rainfall distribution and amount determines the suitability of crops and related agronomic management at different locations (Muthoni et al., 2019). Thus, long-term mean for annual rainfall of the study period (2009 – 2015) were generated and plotted to visualize regional spatial-temporal patterns. The annual spatial temporal variation of rainfall (inter-annual variability) was analyzed by calculating the coefficient of variation (CV) in Eq. (3). Additionally, long-term mean monthly rainfall for the region was analyzed to identify the seasonality of monthly rainfall as the success or failure of the crop is more dependent on seasonal rainfall distribution (Ngetich et al., 2014).

$$CV = \left(\frac{SD}{mean}\right) \times 100 \tag{3}$$

Where; mean and SD are the mean rainfall and standard deviation for a selected temporal scale. According to Asfaw et al., (2018), CV is used to classify the degree in variability of rainfall events as; low (CV < 20%), moderate (20% < CV < 30%) and high (CV > 30%).

## 2.5 Default Model set up

The SWAT+ model (revision 60.5) was set up with the QGIS interface using the data in Table 1 and run for a period of 7 years (2009 – 2015). The harmonized land use product (LUH2; Hurtt et al., 2020) used in this study is formatted as NetCDF, hence the SWAT+ code had to be adapted to include subroutines to read the NetCDF data using and approach proposed by (Chawanda
et al., 2020). The land use map is a composite of land use layers with each layer representing a fraction of a given land use. The fraction layers representing cropland include; C3 annual crops (C3ann), C3 perennial crops (C3per), C4 annual crops (C4ann), C4 perennial crops (C4per) and C3 nitrogen-fixing crops (C3nfx). In the default model setup, the cropland use in the land use map was represented with a uniform generic crop as default in the SWAT+ database (Arnold et al., 2013) for all the heterogenous cropland areas and heat units used to trigger the cropping seasons.
The study area was discretized into 768 landscape units and 12526 unsplit HRUs. The USDA Soil Conservation Service (SCS) curve number method was used to estimate surface runoff, variable storage method selected for flow routing and the Penman-Monteith method (Monteith, 1965) used to calculate the potential evapotranspiration.

## 2.6 Proposed scheduling in the revised SWAT+ model - Crop scheduling with global phenology datasets

In the proposed scheduling, the fraction layers (C3ann, C4per, C4ann, C4per and C3nfx) representing cropland in the land use
map (LUH2) were extracted and comparison was made on a pixel by pixel basis. Whatever crop layer fraction occupied a larger percentage for the rainfed and irrigated agricultural areas within a pixel was selected to represent cropland for irrigated and rainfed areas in that pixel. For example in Figure 2; if the C4ann and C3nfx crop occupied a larger fraction within a pixel compared to other cropland use fraction layers for irrigated and rainfed cropland respectively, they were selected to represent cropland use in that pixel. A crop map was developed from this pixel by pixel analysis and a representative crop selected for
each cropland use fraction based on literature (Leff et al., 2004) as shown in Table 2.

**Table 2:** Representative crop for LUH2 cropland used in SWAT+

| cropland (LUH2) | Representative crop (SWAT+) |
| --- | --- |
| C3 annual | wheat |
| C3 perennial | banana |
| C4 annual | maize |
| C4 perennial | sugarcane |
| C3 nitrogen-fixing | soybean |

For both rainfed and irrigated areas, the representative crops with the corresponding crop phenology (plant and harvest dates) and crop management practices (irrigation, N-fertilizer and P-fertilizer) were extracted from the respective global datasets (Table 1). The extracted data was written in a decision table for each cropland HRU using a python code. The default model
was re-run with the modified crop scheduling with data from global datasets and referred to as 'revised SWAT+' model from here on.

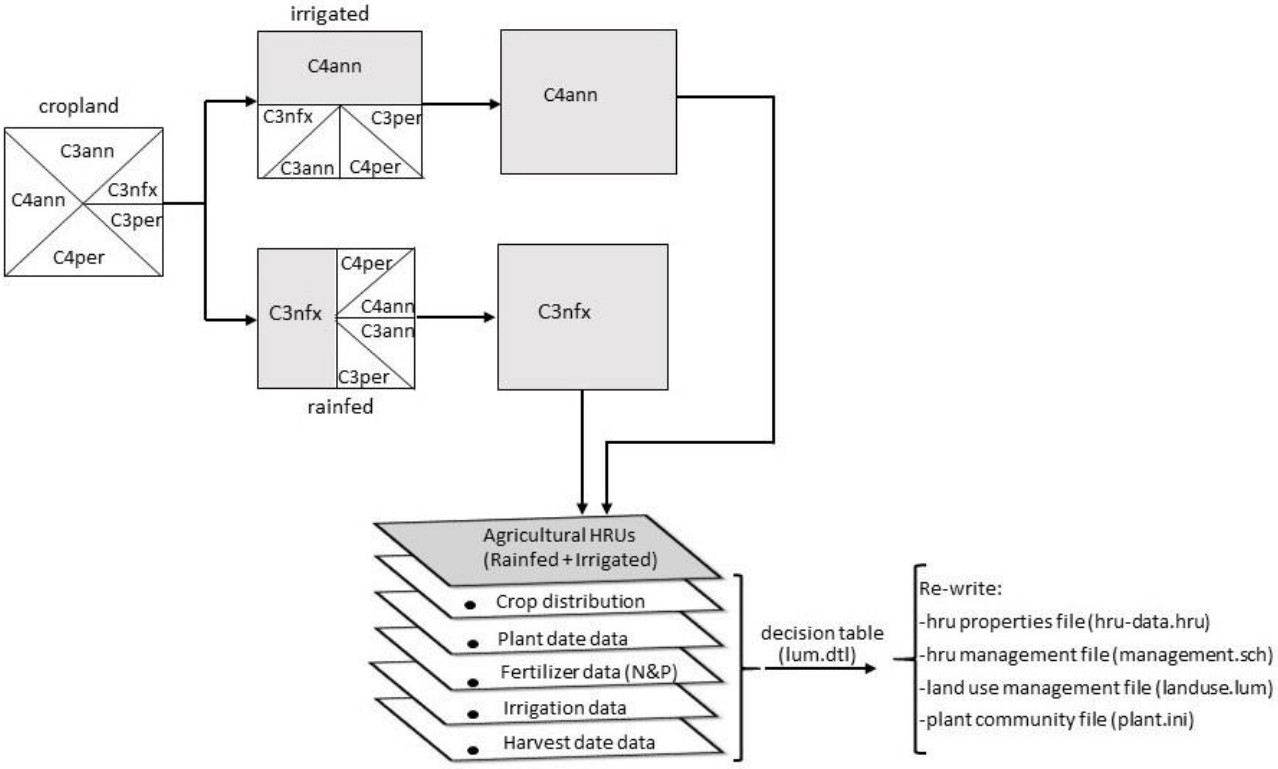

**Figure 2:** Workflow for incorporating crop phenology and crop management data from global datasets into the model

### 2.7 Validation of model results

Our study focused on improved cropland use representation. We evaluated our simulations for LAI and ET for a period of 7 years (2009 – 2015) using remote sensing products in Table 3. Studies (e.g., Alemayehu et al., 2017; Ha et al., 2018; Nkwasa et al., 2020) have demonstrated the capability of using remote sensing products to evaluate hydrological model outputs.

Table 3: Remote sensing datasets used for model evaluation

| Dataset | Resolution | Source |
|---------|-----------|--------|
| LAI | 1 km | CGLS (https://land.copernicus.vgt.vito.be/) |
| ET | 250 m | WaPOR (FAO, 2018) |

Representative basins in the model as shown in Figure 3 were selected to highlight the importance of incorporating global
phenology datasets on LAI simulations in regional hydrological modelling. The selected basins were based on the reported cropping patterns that start with the rainy season (Waha et al., 2013) i.e Upper Blue Nile basin with a predominantly single cropping season, Victoria basin with a double cropping season and the Nile delta with mainly a double irrigated cropping season (Sugita et al., 2017; M. El-Marsafawy et al., 2018). Crop HRUs within the selected sub-basins, that occupied the largest

areas were selected to reduce the effect of mixed LAI from different land cover classes when comparing with the remote

sensing LAI.

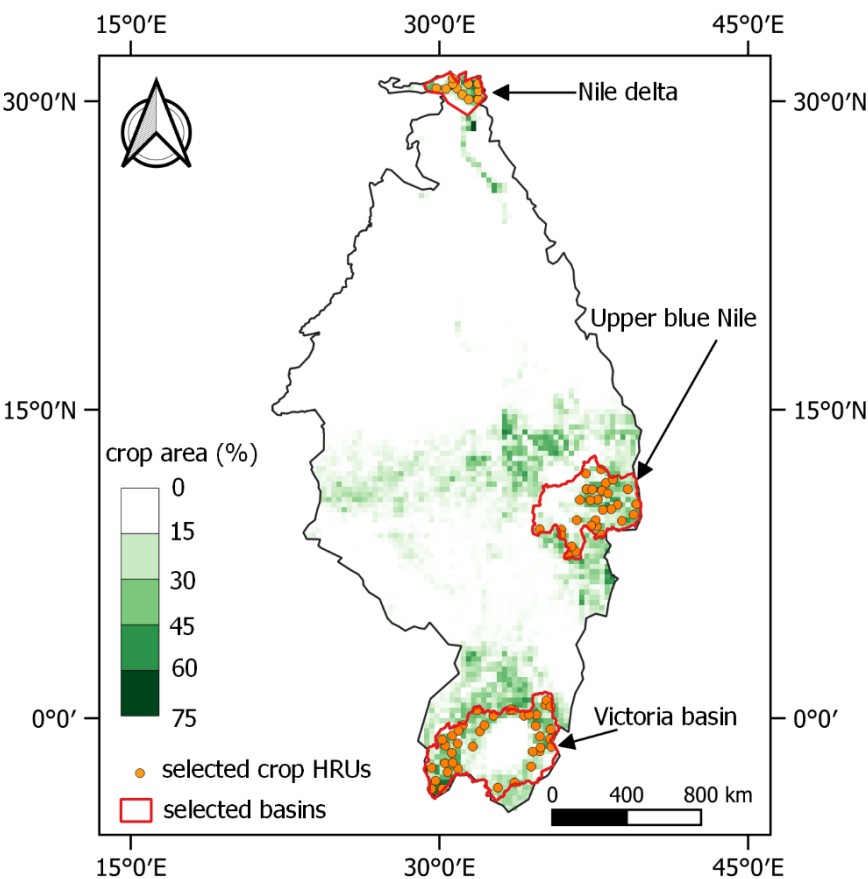

**Figure 3:** Crop area percentage and selected basins for LAI evaluation

Additionally, the correlation coefficient matrix, Eq. (4) was used for model evaluation of LAI.

$$r = \frac{\sum_{i-1}^{n}(Y_{m_i}-\bar{Y}_m)(Y_{o_i}-\bar{Y}_o)}{\sqrt{\sum_{i-1}^{n}(Y_{m_i}-\bar{Y}_m)}\sqrt{\sum_{i-1}^{n}(Y_{o_i}-\bar{Y}_o)}} \qquad (4)$$

where; Where; $Y_{m_i}$ and $Y_{o_i}$ are the simulated and observed values at every time step, with $\bar{Y}_m$ and $\bar{Y}_o$ being the respective mean

values.

To illustrate the impact of revised cropland use representation on model outputs, we compare the differences in soil erosion

simulations between the default and the revised SWAT+ models. However, due to the sparse and poor quality records of

erosion and sediment yield in this region (Haregeweyn et al., 2017), it was not possible to quantitatively validate erosion model

results. Instead, we adopted a 'plausibility check' approach that is suitable for cases when observations for comparison with

model outputs are limited. We compared our erosion estimates for some catchments e.g. Upper Blue Nile with those from a

few previous studies (Hurni, 1985; Betrie et al., 2011; Haregeweyn et al., 2017). Additionally, the improvement in the

representation of crop phenology and crop management practices is expected to minimize errors associated with estimating soil erosion, specifically the crop management factor estimation in the MUSLE.

Both model setups were uncalibrated but checked for the water balance. The study targets improving the default model simulations by better representing the physical land processes (crop growth and ET estimation). Thus, default parameterization was used and we assume that the differences seen in the model setups originate primarily from the crop representation and management practices. This approach could not only isolate the uncertainty in the model due to crop representation but also allow the model results be compared in default parameter conditions, considering parameter calibrations vary with different

catchments. Nkwasa et al. (2020) suggested that improved representation of crop and agricultural land use processes should in fact precede any model calibration efforts. Qi et al. (2020) highlighted the importance of improving process representation in a default SWAT model to ensure reliability of the model in large ungauged basins. Besides, SWAT was developed with the objective of predicting the impact of management on water, sediment and agricultural yields in large 'ungauged' basins (Arnold et al., 1998; Srinivasan et al., 2010). This paper aims for a better physical representation of the land surface processes of the

default model. Hence, we do not address issues concerning the SWAT+ model calibration and validation in this paper.

## 3. Results and discussion

### 3.1 Spatial-temporal variability of rainfall in the region

The long term mean rainfall for 7 years (2009 – 2015) in the region ranged from 0 and 2200 mm (Figure 4 (a)). The highest annual rainfall values are recorded around the equatorial region (Victoria basin) and within parts of Ethiopia (Blue Nile basin).

Most arid areas (parts of Egypt) were the driest regions receiving zero to negligible rainfall within the study period. Figure 4 (b) shows the annual spatial temporal variation using the coefficient of variation (CV) metric. Inter-annual variability was highest ( CV > 50%) in the driest region (parts of Egypt) that coincides with the lowest long term mean rainfall. The rest of the region had low (CV < 20%) to moderate ( 20% < CV < 35%) inter-annual variability of rainfall which means that in most parts of the region, the total annual rainfall remained relatively stable. A study by Muthoni et al. (2019) in East Africa also

reported relatively low inter-annual variability ( < 10 %) with in the Victoria basin.

Figure 5 shows the long-term mean monthly rainfall pattern for selected HRUs in the selected basins (Figure 3) in the region. The rainfall in the Victoria basin located within the equatorial region exhibits a bimodal pattern with the main wet season in March – May and a short rainy season in October – December. In the upper blue Nile basin located within Ethiopia, there is only one main wet season in the months of June – September. For the Nile delta in Egypt, it is seen that the wet seasons occur

in March – May and October – February with values far lower than the other regions. These monthly patterns are consistent to those indicated by Onyutha and Willems (2015). The wet seasons (start and end of rainy period) represent the major cropping seasons in the Nile basin as rainfall is the primary controlling factor for leafing and senescence in tropical and subtropical regions (Ma et al., 2019).

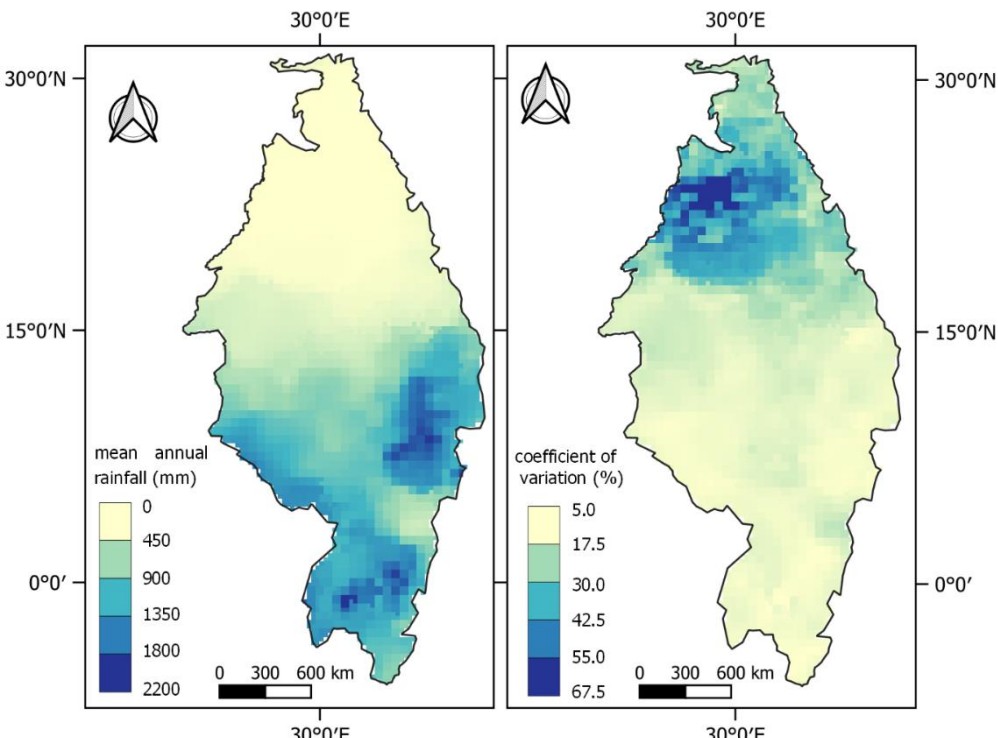

**Figure 4:** (a) Long term mean annual rainfall (2009 - 2015); (b) Coefficient of variation of annual rainfall

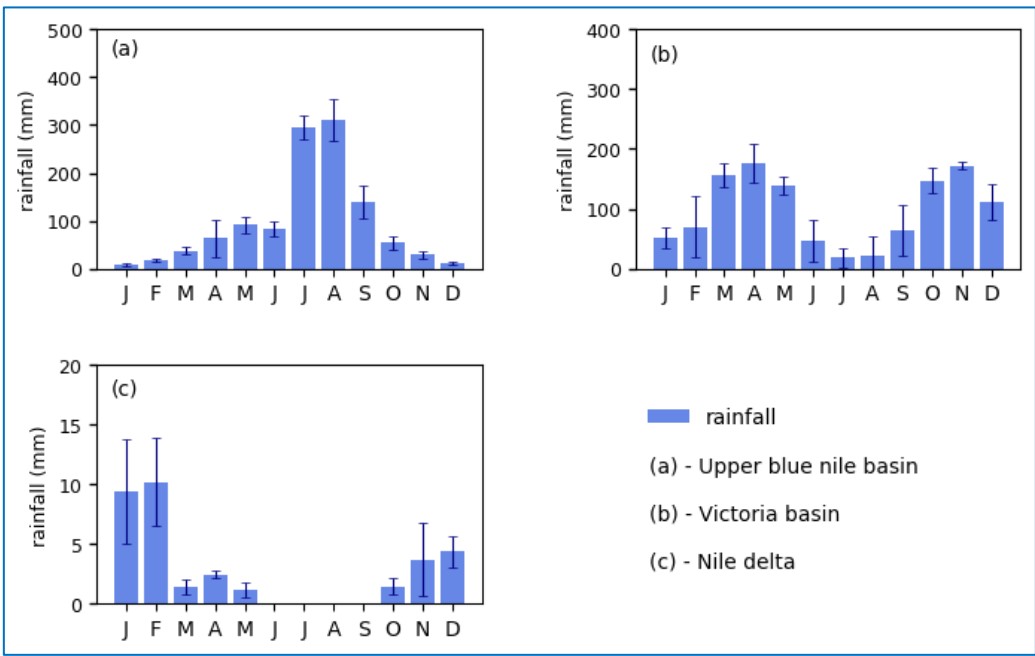

**Figure 5:** Long term monthly mean and standard deviation (lines) rainfall pattern for selected HRUs in selected basins within the region

## 3.1 LAI simulations

The simulated LAI from both the default and revised SWAT+ models was compared with the remote sensing LAI extracted for the maize, wheat and soy HRUs in the 3 selected sub-basins (Upper Blue Nile, Lake Victoria and Nile Delta). In the Upper Blue basin, Figure 6 (a) and Figure 6 (c), there is an improved LAI simulation in the revised SWAT+ model with the phenological development being captured in the correct major cropping season within the rainy season for both the rainfed and irrigated maize HRUs. Additionally, the revised SWAT+ model LAI strongly correlates ($r_d > 0.5$) with the remote sensing

(RS) LAI. Figure A1(a) and Figure A1(c) in the Appendix A, also shows the improvement in LAI simulations for rainfed and irrigated wheat HRUs in the Upper blue Nile basin.

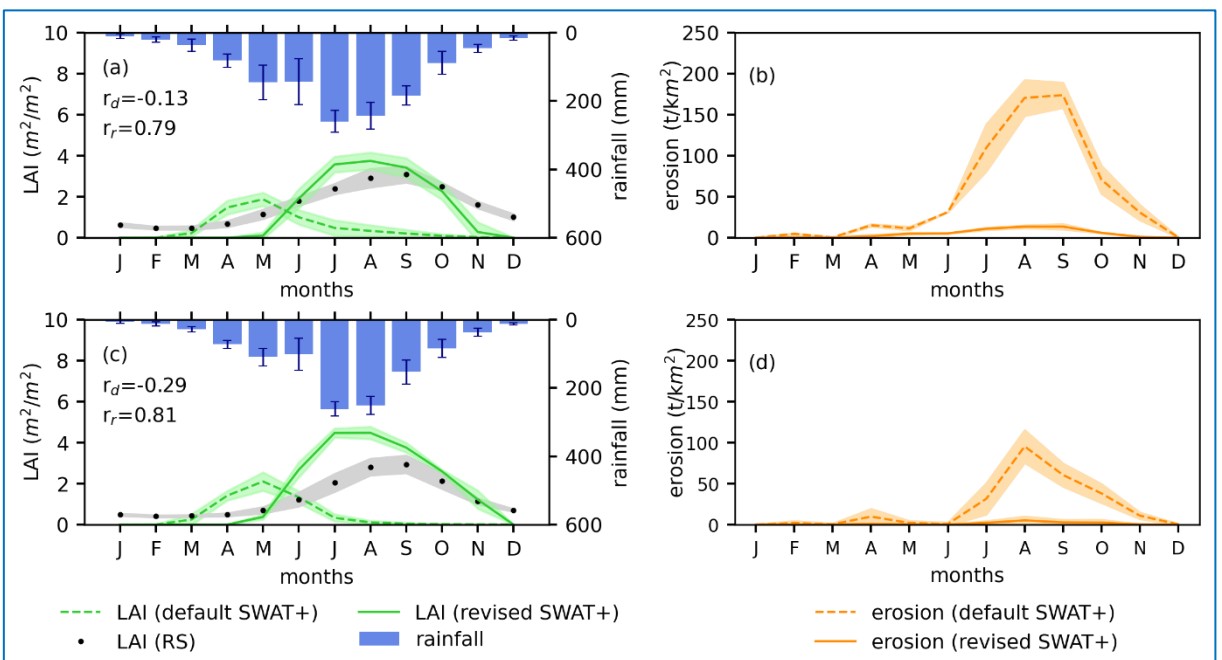

**Figure 6:** Monthly mean and standard deviation (bands) of LAI (a) and erosion (b) comparison for rainfed maize; Monthly mean and standard deviation (bands) of LAI (c) and erosion (d) comparison for irrigated maize; in the Upper Blue Nile basin.

The LAI coefficients ($r_d$ for the default SWAT+ model and $r_r$ for the revised SWAT+ model)

In the Victoria basin, (Figure 7(a) and Figure 7(c)), the revised SWAT+ model captures only one cropping season in comparison to the RS LAI that shows a double seasonal pattern agreeing with the rainfall. This is because the global data set (Jägermeyr et al., 2021) utilized captures only the main cropping season per pixel per crop and hence the model misses the additional cropping seasons. Additionally, Figure A2 (a) and Figure A2 (c) in the Appendix A also show a single cropping

season captured in the Victoria basin for irrigated maize HRUs with some HRUs having a cropping season from April to November (irrigation in the dry season) while others have a cropping season from September to January (irrigation in the second rainy season). There is also a slight improvement in the LAI correlations for the default and revised SWAT+ models

with RS LAI in the Victoria basin as the LAI simulated by the revised SWAT+ model are indicative of the representative crops planted in the basin as compared the generalized crop representation in the default model.

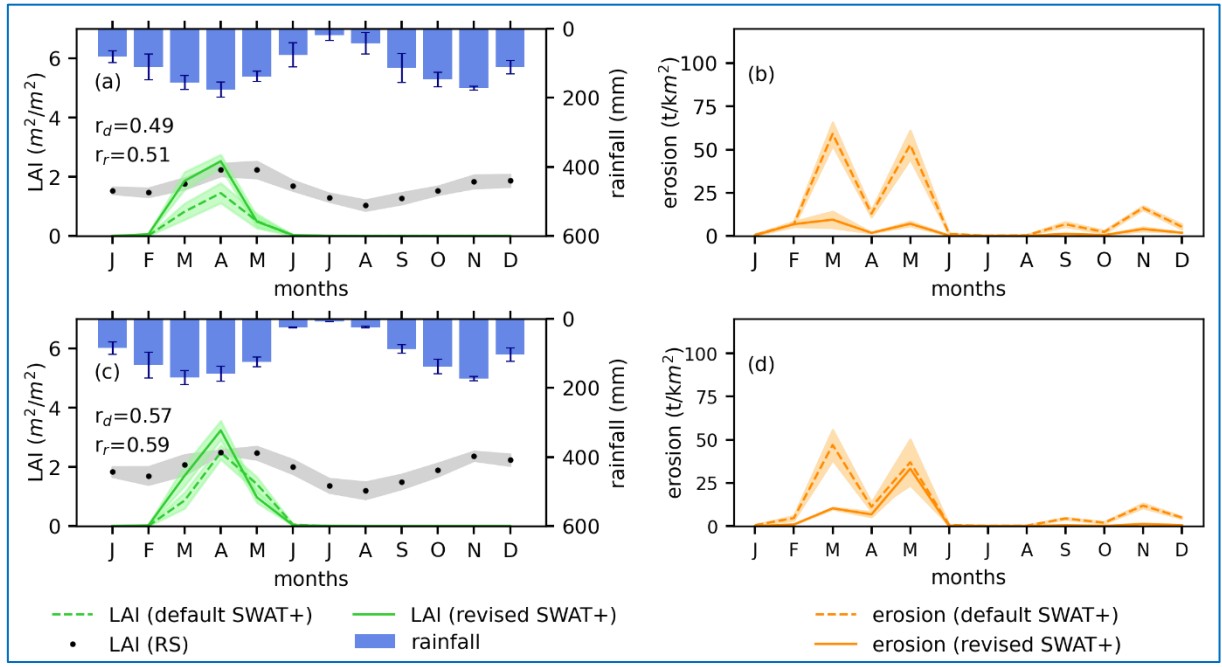

**Figure 7:** Monthly mean and standard deviation (bands) of LAI (a) and erosion (b) comparison for rainfed wheat; Monthly mean and standard deviation (bands) of LAI (a) and erosion (b) comparison for irrigated wheat; in the Victoria basin. The LAI correlation coefficients ($r_d$ for the default SWAT+ model and $r_r$ for the revised SWAT+ model)

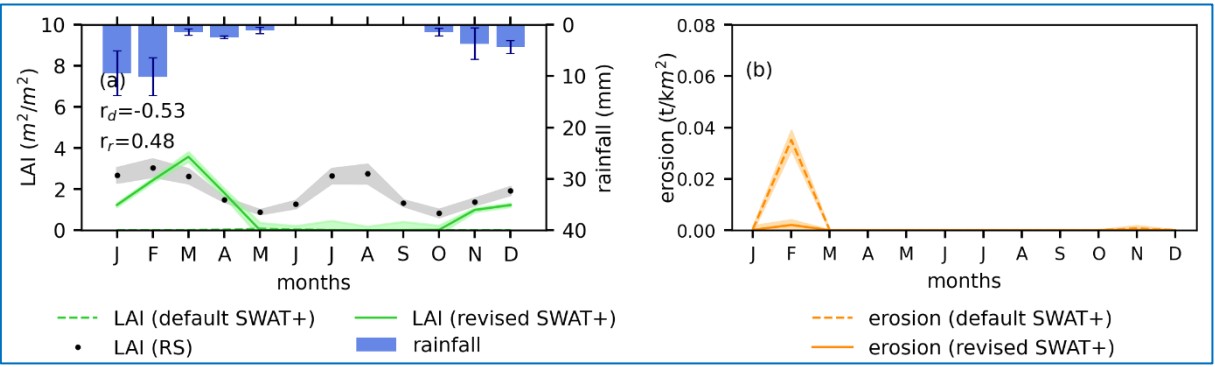

**Figure 8:** Monthly mean and standard deviation (bands) of LAI (a) and erosion (b) comparison for irrigated wheat; in the Nile delta. The LAI correlation coefficients ($r_d$ for the default SWAT+ model and $r_r$ for the revised SWAT+ model)

For the Nile delta that is predominantly irrigated, the revised SWAT+ model improves the LAI simulations (from $r_d$ = -0.53 to 0.48) as compared to the default SWAT+ model that simulates a negligible LAI, Figure 8 (a). Without management practices (irrigation and fertilization), plant growth in the default SWAT+ model is constrained in the Nile delta being a predominantly dry region resulting into low LAI simulations. However, with the cropping calendar and the associated management practices,

LAI simulations are improved in the revised model. The revised SWAT+ model still captures only one cropping season as compared to the RS LAI that shows two cropping seasons that are also highlighted in previous studies (M. El-Marsafawy et al., 2018).

The interannual variability of LAI within the selected basins was also examined for selected crops in Figure 9, but no significant interannual variations were noticed. This can be explained by the low interannual variability of rainfall in most parts of the region (Figure 4 (b)) within the study period. However, the seasonal patterns remained consistent with the LAI peaking in the wet/rainy seasons within all the selected basins. The low interannual variability of LAI within the study period certainly does not imply that the relevance of the variability of LAI interannually is unimportant in the region as an analysis on a longer timeseries could yield different results.

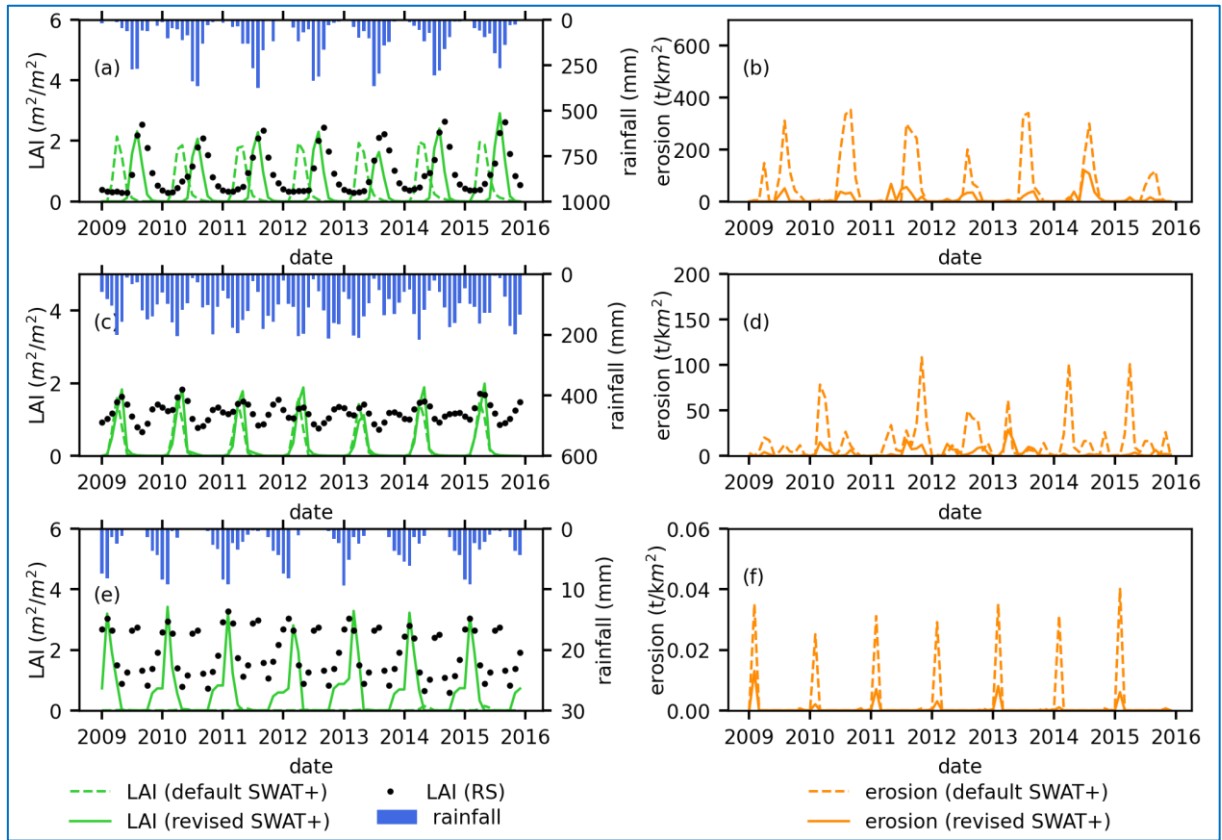

**Figure 9:** Monthly LAI comparison for rainfed maize (b); Monthly erosion estimates for rainfed maize in Upper blue Nile basin; (c) Monthly LAI comparison for rainfed wheat; (d) Monthly erosion estimates for rainfed wheat in the Victoria basin; (e) Monthly LAI comparison for irrigated wheat; (f) Monthly erosion estimates for irrigated wheat in the Nile delta

From all the basins, we see an improved seasonal temporal crop-growth phenological development pattern with the revised model as compared to the default model for both the rainfed and irrigated regions which underscores the relevance of the methodological advancements in this paper. However, in the Victoria basin and the Nile delta where we have two dominant cropping seasons, the global datasets still capture one cropping season. Additionally, some regions in East Africa have been

reported to have up to 3 cropping seasons (Waha et al., 2013; Msigwa et al., 2019) which have not been captured in these simulations. The global crop calendars also lack a temporal time series dimension which could be a substantial source of

uncertainties in predicting phenological events of croplands. The lack of observational data of multiple cropping seasons at regional scale has been reported in previous studies (Rounsevell et al., 2003). Some studies (Ma et al., 2019; Rajib et al., 2020) have used remote sensing LAI datasets e.g. MODIS to improve LAI simulations with SWAT in tropical sub-tropical regions. Remote sensing can be useful for characterization of cropping systems, however expert judgment and local knowledge is still required for crop type and crop management mapping (Bégué et al., 2018). Combining remote sensing datasets with existing

global phenology datasets provides potential for addressing the gap in multiple cropping datasets. With recent research progress in cropping patterns such as the crop generator (Sietz et al., 2021), used to reproduce crop rotation characteristics at regional scale, the global phenology datasets can be improved to consider multiple cropping seasons.

The use of remote sensing LAI data (1km resolution) in evaluation could also present uncertainties since the remote sensing data does not represent a pure signal of a crop but rather vegetation with in the pixel. Studies (Ma et al., 2019; Nkwasa et al.,

2020) have highlighted these scaling issues when using remote sensing products in model evaluation. Remote sensing pixels are usually presented in a grid system that cannot sufficiently capture spatial details of mixed vegetation within a grid. Nevertheless, the remote sensing data still provides insights on the temporal vegetation growth relationship with seasonal weather patterns.

### 3.2 ET simulations

The annual average simulated agricultural ET from the revised SWAT+ model improves the default agricultural ET simulation from 732 mm y$^{-1}$ to 837 mm y$^{-1}$ as compared to the WaPOR agricultural ET of 936 mm y$^{-1}$. The improvement in the spatial distribution of the agricultural ET is shown in Figure 10. Figures 10 (a) and 10 (b) show the default model simulation and the revised SWAT+ model simulation respectively in reference to WaPOR ET (Figure 10 (c)). The inclusion of the global phenology and management practices shows that ET is one of the major components of a basin water balance that is greatly

influenced by the seasonal vegetation growth cycles. This can be attributed to the improved temporal patterns of LAI which favours transpiration and evaporation from canopy intercepted water. According to Wang et al. (2014), incorporating LAI and vegetation growing stages in modelling could explain half the variability in transpiration to ET ratios across ecosystems. Hence, overlooking crop representation in hydrological models is not physically meaningful because a poor simulation of LAI has a cascading effect on how the model partitions the ET fluxes.

Although, the agricultural ET is improved with the incorporation of the global crop phenology, there is still an underestimation especially in the Nile delta and the equatorial region Figure 10 (d). This underestimation could be mainly attributed to the missing multiple cropping seasons especially in areas that are irrigated such as the Nile delta. As mentioned in the previous section, the phenology datasets give only one cropping season which misrepresents areas with multiple cropping seasons. Furthermore, automatic irrigation was specified in the model, which applies water from a deep aquifer in all irrigation fields

when the water stress is below a specified threshold (0.7) of the field capacity. However, extracting irrigation water from a

deep aquifer at all irrigation fields may be unrealistic, causing uncertainties in irrigation applications which affects the ET estimates.

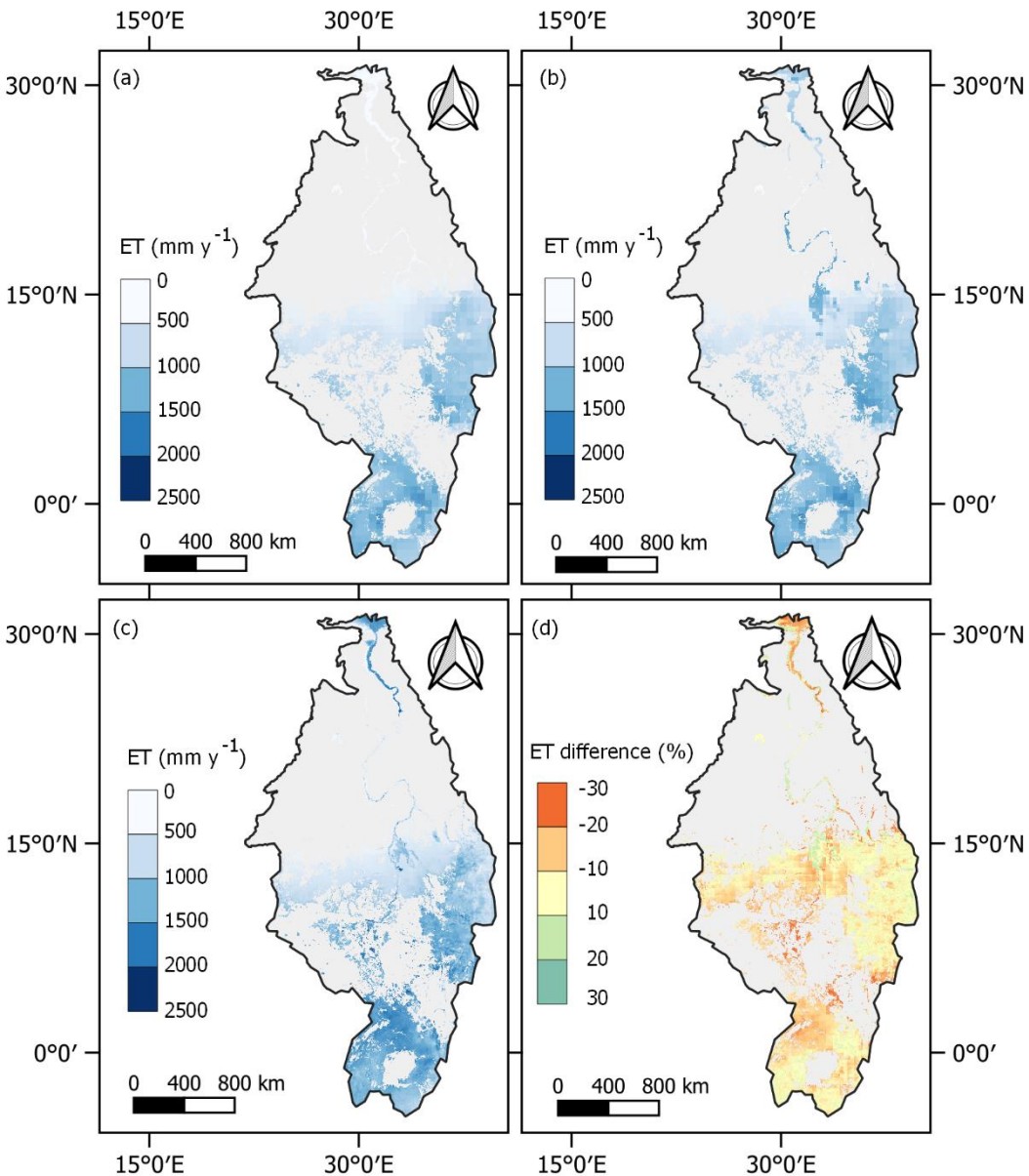

**Figure 10:** Spatial distribution of agricultural ET; a) default SWAT+ model ET, b) revised SWAT+ model ET, c) WaPOR ET, d) ET difference (WaPOR ET – revised SWAT+ model ET)

It is also important to point out that the simplifications made in using a representative crop for the whole pixel could have effects on the ET fluxes due to the simplifications in the variations of the physical characteristics (e.g. LAI, root depth, stomata conductance) of the heterogenous crops (Burakowski et al., 2018). This simplification can also alter the partitioning of sensible

heat fluxes to latent heat fluxes (Eltahir, 1998) that in turn affect the ET estimates. Therefore, at local scales, the heterogeneity

of crops within a pixel should be considered. The default crop parametrization could also be an extra source of uncertainty in the ET estimates. Although these uncertainties exist, incorporation of agricultural land use and the corresponding management practices in hydrological models provides a promising way to improve ET estimates especially for cultivated regions. Additionally, the ET estimates could be further improved by model calibrations to obtain the optimal possible ET.

## 3.3 Erosion simulations

LAI is not only directly related to processes such as rainfall interception, evaporation, transpiration, soil evaporation, root depth but also to soil erosion through canopy cover which varies during the growth cycle of the plant. With a better representation of the cropping season, the rainfall season also corresponds with higher LAI values which results in lower erosion yields.

Figures; 6(b), 6(d), 1A(b), A2(b), A3(d), 4A(b) and A4(d), reveal that the soil erosion estimates are reduced in the revised

SWAT+ model because the canopy cover grows in the correct cropping season (rainy season) reducing the effective energy of intercepted raindrops. These results conform to a study done by Zhao et al., (2013), which showed a strong correlation in soil erosion reduction with the crop growth cycle. In Figure 7 (b) and Figure 7 (d), even though the cropping season in the revised SWAT+ model captures only one cropping season as the default model, there is still a reduction in the HRU erosion estimates because the revised SWAT+ LAI, representative of an actual crop is greater than the default LAI representative of a generic

crop. Hence, we notice that a slight increase in the LAI magnitude has a strong impact on the erosion simulations. Additionally, with a slightly higher LAI magnitude in the revised SWAT+ model, more biomass is generated which results in more residue that could be more effective in reducing soil erosion even after the cropping season. Residue intercepts rain droplets near the soil surface that droplets regain no fall velocity. Thus, a given percentage of residue is more effective than the same percentage of canopy cover (Neitsch et al., 2011). However in Figures 7 (b) and 7 (d), the erosion peaks in the default model are strong

even though the LAI is relatively high. This can be attributed to the reduction in the residue on the soil surface during the second rainy season that occurs with no crop cover. For the Nile delta in Figure 8 (b), the soil erosion estimates reduced further even though they were already insignificant. From Figures; 9 (b), 9 (d) and 9 (f), it is important to highlight that the low to moderate interannual variability of rainfall (Figure 4 (b)) in most agricultural areas of the region, coupled with low interannual variation in the simulated LAI shown in Figures; 9(a), 9 (c) and 9 (e) resulted in low interannual variability of erosion

simulations.

The average annual soil erosion estimates are reduced by a maximum of 625 t km$^{-2}$ y$^{-1}$ (mostly in the upper blue Nile basin), Figure 11 (a) or up to 90 %, Figure 11 (b) in some areas within the region when using the revised SWAT+ model as compared to the default model. The average regional soil erosion yield reduced by 16 % with the most decrease of 37% in the Upper blue Nile basin. This reduction is attributed to the improved timing of the cropping seasons in correspondence to the start of

the rainy season which provides more canopy cover to intercept the raindrops. However, in some isolated regions, the revised SWAT+ model simulated an increase in soil erosion estimates as compared to the default model. In most of those regions, the

global phenology data captures the irrigated cropping season which is often occurring in the dry seasons (Figure A3(a)) which causes discrepancies by not representing the major growing season in the rainy season. As mentioned in the previous section, this is attributed to the fact the global phenology data provides a single cropping season per pixel per year.

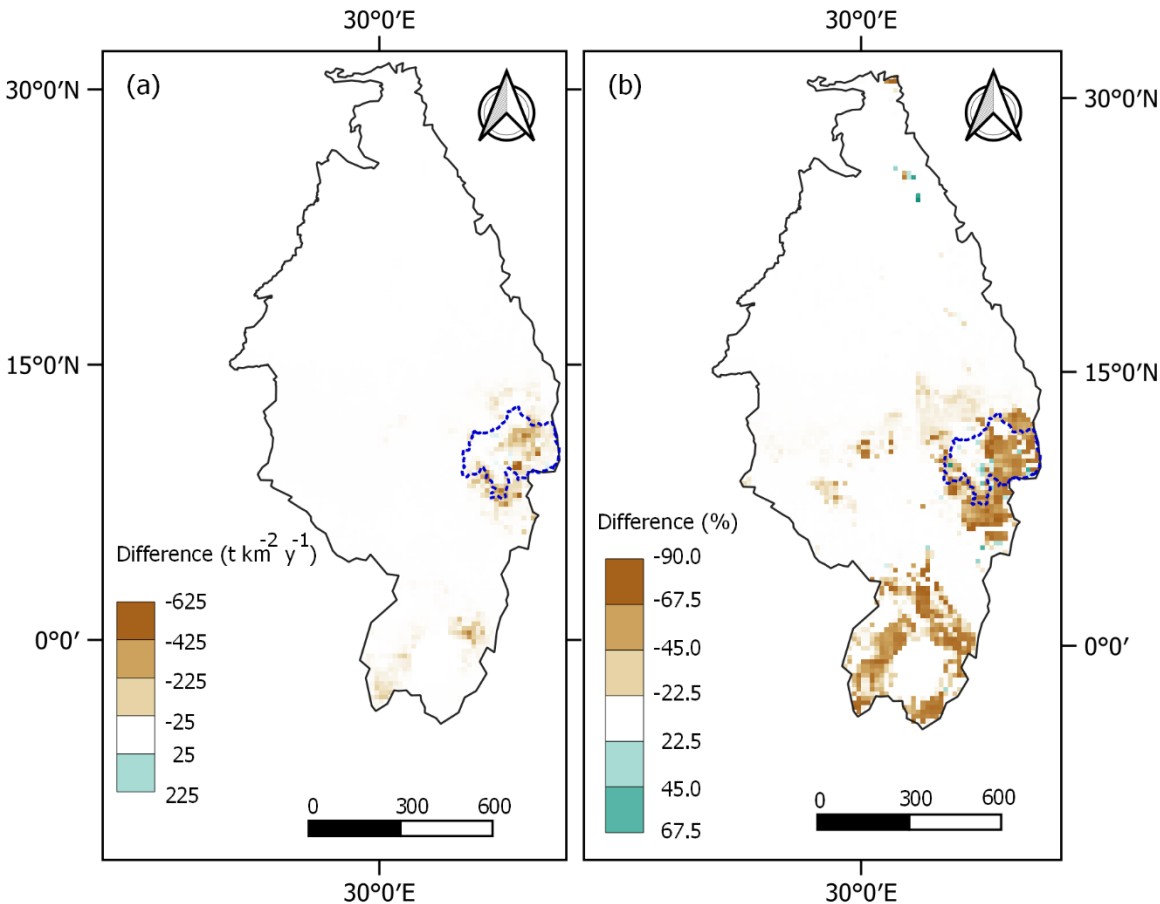

**Figure 11:** Change in erosion estimates (revised SWAT+ model minus default SWAT+ model); (a) absolute differences, (b) percentage differences

Among the few hydrological model applications in the subtropics that focus on improved erosion simulations, Ma et al., (2019) enhanced the SWAT model performance by using remote sensed LAI to give reasonable crop cover estimates, leading to an accurate estimate of soil erosion and sediment yield. However, when using remote sensing data, detecting crop types and cropping sequences without local knowledge or ground truth data is not possible (Bégué et al., 2018), which emphasizes the importance of the approach proposed in this study.

In order to validate the regional soil erosion estimates, the simulated soil loss from the revised SWAT+ model was compared with the spatial patterns in erosion rates from the literature. From published literature, Ethiopia is the one of the most documented countries in Northeast Africa with marginal information existing for other countries (Haregeweyn et al., 2015). The revised SWAT+ model shows that the regional soil erosion extent varies from zero to over 20500 t km$^{-2}$ y$^{-1}$, (Figure 12),

revealing the severity of soil erosion in the Ethiopian highlands as compared to the other parts of the region. Ethiopian highlands have been reported to have high soil erosion and sediment yield rates attributed to partly topography and rainfall but also to recent and historic land conversions to agriculture.

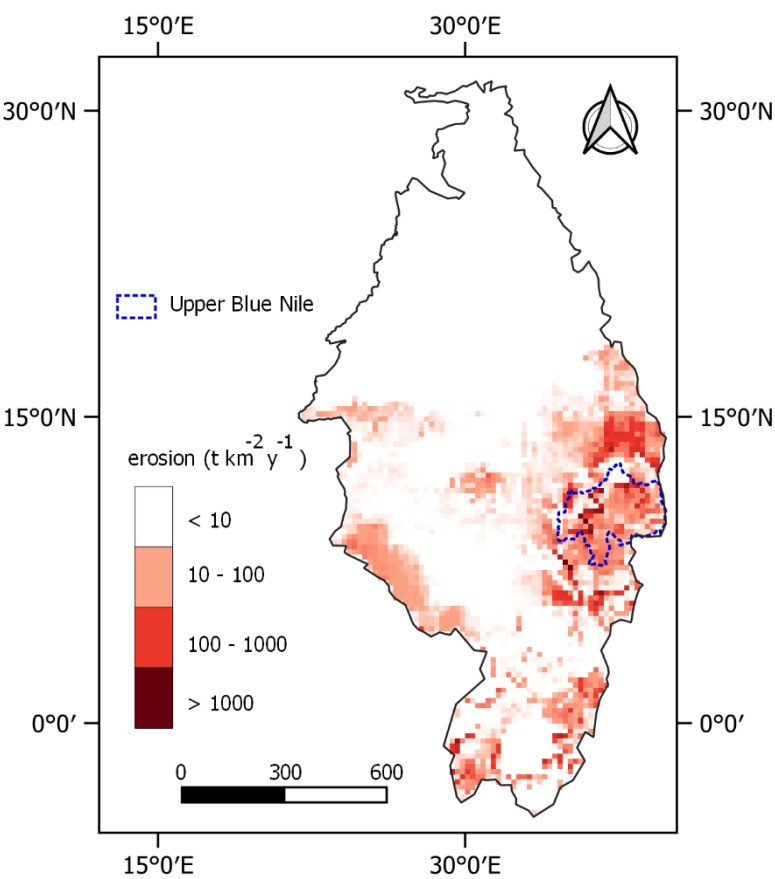


**Figure 12:** Spatial distribution of predicted annual average soil erosion at HRU level in Northeast Africa (2009 – 2015)

Comparing with estimates from the Upper Blue Nile basin, the model estimated an erosion yield extent from 0 to 13000 t km$^{-2}$ y$^{-1}$ and a mean of 701 t km$^{-2}$ y$^{-1}$ which is slightly lower but comparable to a net soil erosion mean of 734 t km$^{-2}$ y$^{-1}$ reported by Haregeweyn et al., (2017) and soil erosion yield extents from zero to over 15000 t km$^{-2}$ y$^{-1}$ reported by Hurni, (1985), Betrie et al., (2011) and Haregeweyn et al., (2017). Tamene and Le (2015) reported a net soil loss of 8500 t km$^{-2}$ y$^{-1}$ and 600 t km$^{-2}$ y$^{-1}$ in the Blue Nile and White Nile basins respectively. These estimates should be considered as indicative as comparing these values with the Northeast African regional model estimates can be challenging mainly due to the differences in the sizes of areas involved resulting from the different delineation procedures.

Even though the regional model underestimates the soil erosion in comparison with these localized studies, the order of magnitude is within the same range. The underestimation can be attributed to the finer resolution of datasets utilized by the local studies as compared to the coarse datasets utilized in the regional model. For example, Molnár and Julien, (1998)

calculated soil erosion using different DEM grid sizes and concluded that the estimated slope gradients decreased as the cell size increased which influenced the topographic factor (LS) estimation. Additionally, the input global weather data is at a scale of 0.5º which makes it too coarse to capture the spatial variability of weather at a finer scale. This has been a challenge for large scale hydrological modelling (Chawanda et al., 2020), that needs to be addressed for better performance.

With that background, it is not wise to entirely consider the soil erosion estimates in this study as exact quantification but rather as close approximations. It is worth noting that the focus of this study was not soil erosion estimation but to illustrate a concept.

## 5. Conclusion and recommendation

In this work, an approach has been developed for an improved representation of crop phenology and management in a regional SWAT+ model using decision tables and global datasets. In addition, global remote sensing datasets of LAI and ET have been used for model evaluation. A comparison of the simulated LAI revealed improved temporal growth patterns in agreement with remote sensing LAI, especially for regions with a single cropping cycle. However, for regions with multiple cropping cycles, only one cropping cycle was represented as most global phenology datasets provide a single cropping cycle per year.

The improvements in the SWAT+ model reduced the agricultural ET deficit by 50 % in comparison with the WaPOR ET, showing a strong linkage between hydrological response and agricultural land use representation. Additionally, this improvement in ET estimates is expected to reduce any calibration efforts needed to obtain the maximum possible ET as the physical process representation of crops is improved. A considerable reduction of 16 % in the average regional soil erosion estimates was noticed after implementing this approach. This impact on soil erosion estimates shows the importance of proper representation of crop processes and an important element for minimizing errors in soil erosion estimates. These findings emphasize the importance of advancing process representation in physically-based models to improve the model reliability.

There is a need for global phenology datasets with multiple cropping seasons for further improvements in the crop representation, especially for improving crop processes in irrigated areas or areas with multiple rainy seasons. For example; mapped global areas of different multiple cropping systems (Waha et al., 2020), can potentially be combined with global phenology datasets to generate a global crop calendar with different cropping systems. The approach developed in this research lays a foundation for improved agricultural land use representation with associated management practices at regional and global scales which will further improve regional to large scale hydrological and water quality impact assessments of global change.

**Appendix A**

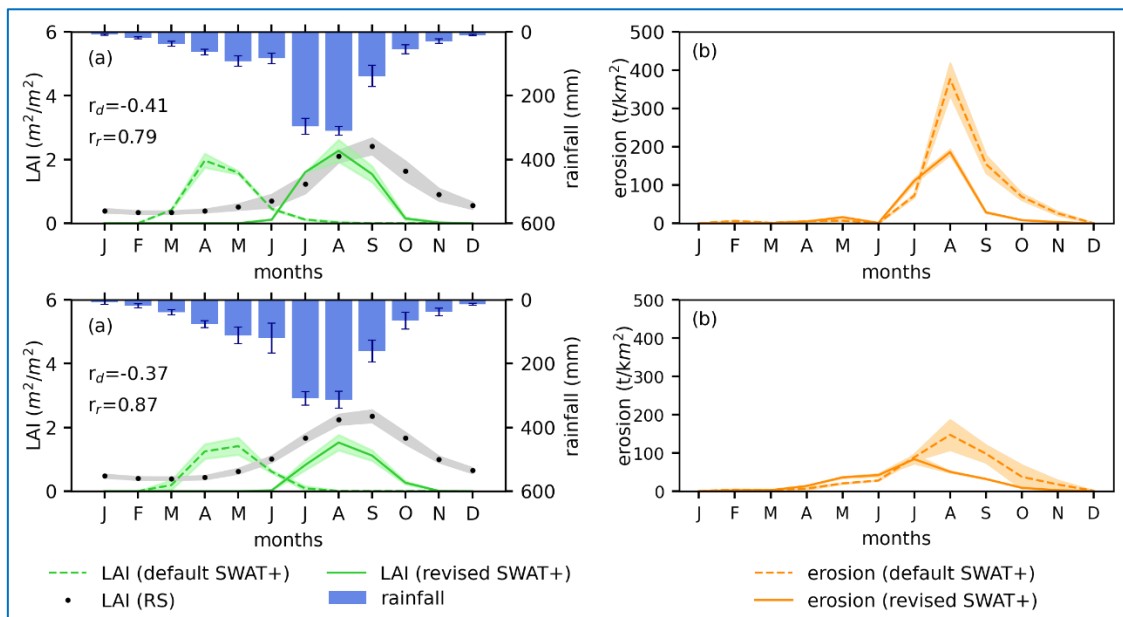

**Figure A1:** (a) Monthly mean and standard deviation (bands) of (a) LAI and (b) erosion comparison for rainfed wheat; Monthly mean and standard deviation (bands) of (c) LAI and (d) erosion comparison for irrigated wheat; in the Upper Blue Nile basin. The LAI correlation coefficients ($r_d$ for the default SWAT+ model and $r_r$ for the revised SWAT+ model)

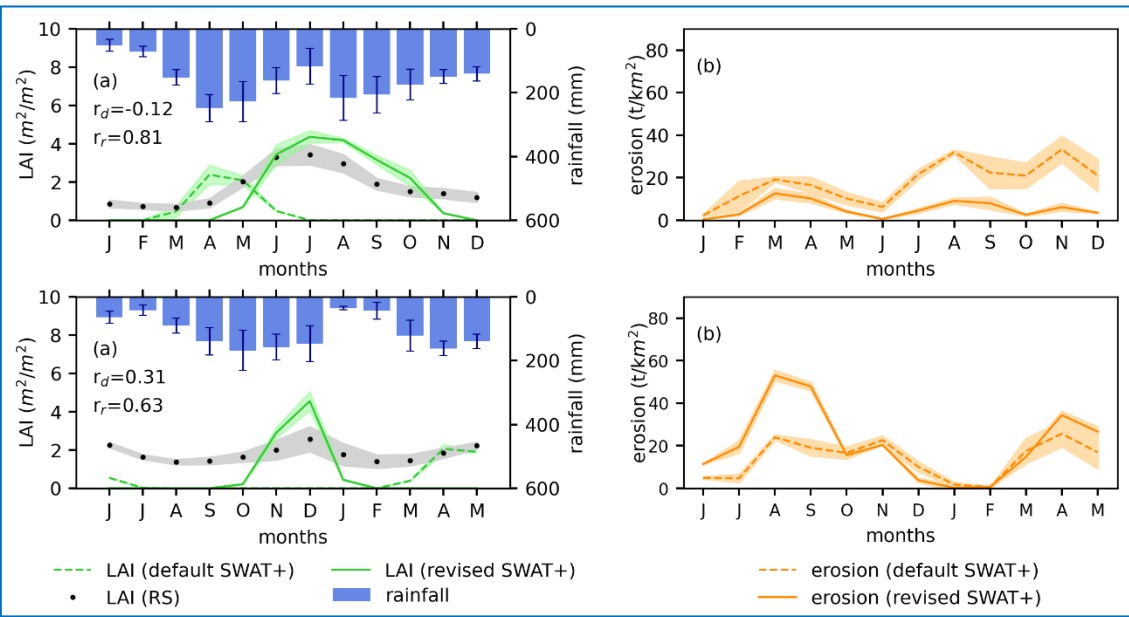

**Figure A2:** Monthly mean and standard deviation (bands) of (a) LAI and (b) erosion comparison for irrigated maize – case1 (irrigation during the dry growing season); Monthly mean and standard deviation (bands) of (c) LAI and (d) erosion comparison for irrigated maize – case2 (irrigation during the main wet growing season); in Victoria basin. The LAI correlation coefficients ($r_d$ for the default SWAT+ model and $r_r$ for the revised SWAT+ model)

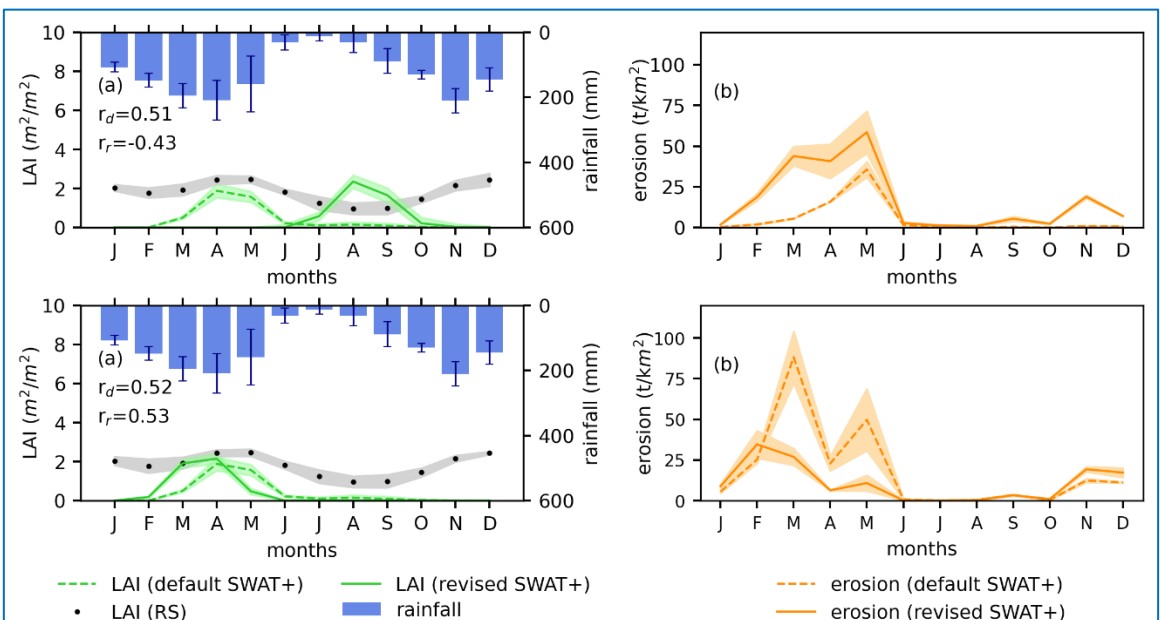

**Figure A3:** Monthly mean and standard deviation (bands) of (a) LAI and (b) erosion comparison for irrigated soy - case1 (irrigation during the dry growing season); Monthly mean and standard deviation (bands) of (c) LAI and (d) erosion comparison for irrigated soy – case2 (irrigation during the main wet growing season); in Victoria basin. The LAI correlation coefficients ($r_d$ for the default SWAT+ model and $r_r$ for the revised SWAT+ model)

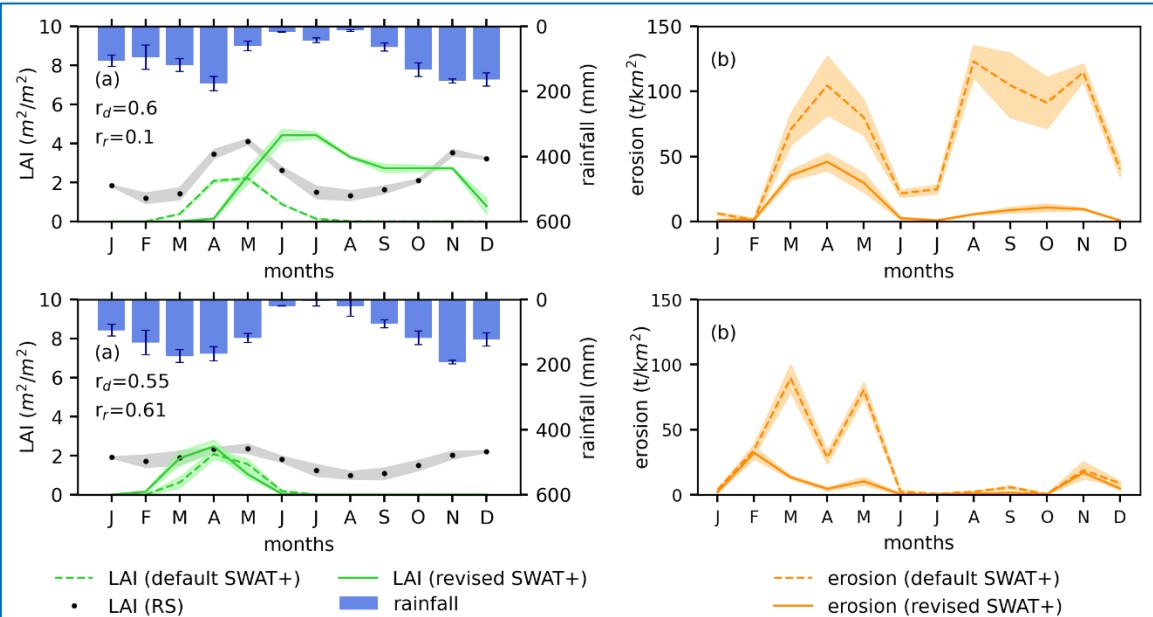

**Figure A4:** (a) Monthly mean and standard deviation (bands) of (a) LAI and (b) erosion comparison for rainfed maize; Monthly mean and standard deviation (bands) of (a) LAI and (b) erosion comparison for rainfed soy; in the Victoria basin. The LAI correlation coefficients ($r_d$ for the default SWAT+ model and $r_r$ for the revised SWAT+ model)

**Code availability:** This approach was created using python scripts available on the VUB-HYDR repository (https://github.com/VUB-HYDR/2021_Nkwasa_etal_HESS). The revisions of the scripts are managed there and are available
on request.

**Author contributions:** AN and AG designed this study. JJ provided the phenology datasets. CJC and AN set up the model. AN performed the model simulations, primary analysis and drafted the paper. All authors contributed to results interpretation and reviewed the paper.

**Competing Interests:** The authors declare that they have no conflict of interest.

**Acknowledgement:** The authors thank the Research Foundation – Flanders (FWO) for funding the International Coordination Action (ICA) "Open Water Network: Open Data and Software tools for water resources management" (project code G0E2621N) and the Flemish Research Council (VLIR) for funding the JOINT project "Global Open Water Academic Network: Joint Research and Education on Open Source Software for Integrated Water Resources Management" (project code TZ2019JOI022A105) and the EU H2020 programme for funding "Water-ForCE – Water scenarios For Copernicus
Exploitation" (grant agreement No. 101004186).

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
