# Peer review of "Improved Representation of Agricultural Land Use and Crop Management for Large Scale Hydrological Impact Simulation in Africa using SWAT+"

_Hydrology and Earth System Sciences, 2021_

## Author Comment (AC1)

**Reviewer 1**

The paper presents an important step forward to better capture crop phenology in hydrological modelling. It is well written and structured. To demonstrate its reach and inspiration for future modelling, the paper could be better positioned with regard to existing modelling advances considering crop dynamics. It would gain from a discussion of results in view of existing modelling approaches and advances, going beyond the mention of omitting these in most models. Discussing uncertainties arising from the sole focus of the most widespread crop per pixel and simulation of the 5 representative crops would also make the paper stronger.

**Response:** Thank you for a positive evaluation of our study and your time to provide critical comments to improve the manuscript.

A discussion of results (LAI and ET) in view of existing approaches is to be added in the revised manuscript. Uncertainties arising from sole focus on the widespread crop per pixel as compared to the simulation of the 5 representative crops is to be added under the discussion of ET simulations. This is because the heterogenous representation of crops in a pixel can have effects on the ET fluxes due to variations in the physical characteristics of vegetation such as root depth, LAI, stomata conductance, surface roughness and albedo (Burakowski et al., 2018).

**Further remarks**

- The start of the introduction would be more interesting to read if it wasn't a copy of the abstract.

**Response:** Yes, this is to be rectified in the revised manuscript.

- Lines 34-35: some details of the "simplistic way"/"abstract terms" of considering crops in hydrological models and what we learn from these would provide a good start.

**Response:** Lines 35 – 37 give some details of the "simplistic way/abstract terms" of considering crops in hydrological models. Specifically, by highlighting how most models neither address crop phenological development nor distinguish between different crops and the associated management practices (e.g planting, irrigation, fertilization, harvesting). We will add information in the revised manuscript about the regional model applications that have considered one uniform generic crop for agricultural land use modelling despite the wide variety of crops existing in agricultural land use.

Lines 37 – 38 gives some information about how cropland representation and the timing of applied management practices in hydrological models affects the water balance components. However, we will add more information specifically on how the efficient Evapotranspiration (ET) estimation is a challenge in agricultural dominated catchments without representation of various crops and crop types. Moreover, using a generic uniform crop for agricultural land use modelling fails to account for any variability in Leaf Area Index corresponding to real life crop scenarios.

- Line 62: examples on current knowledge derived from the implementation of cropping dynamics in hydrological modelling would be useful (e.g. The Crop Generator: Implementing crop rotations to effectively advance eco-hydrological modelling. https://www.sciencedirect.com/science/article/abs/pii/S0308521X21001360).

**Response:** Yes, examples on current knowledge about approaches on representation of crop dynamics in other eco-hydrological models is to be added in the revised manuscript to highlight what we can learn from these approaches.

- Lines 70-75: How is phenological crop development simulated in other hydrological models, e.g. SWIM? Has there been any coupling attempts with modelling of land surface processes that consider phenology, e.g. PROMET?

**Response:** Most hydrological models e.g SWAT+, SWIM have an integration of a hydrologic module and crop growth module (mostly EPIC, Williams and Singh, 1995) and by default, crop growth is simulated by accumulated heat units. In most applications, modelers uses a generic or a constant crop for representation of agricultural land use. As highlighted, the challenge of this representation is twofold; in tropical and subtropical regions, the use of heat units easily causes incorrect cropping seasons as these regions are primarily driven by water availability. Secondly, the use of a generic crop fails to account for any variability in existing heterogenous agricultural land use. A review of coupled hydrologic and crop growth models has been done by Siad et al., (2019).

Coupling attempts of land surface process models e.g PROMET to consider phenology have been done through an agricultural management subroutine which also initializes management actions that may be of hydrologic, climatic or agricultural consequence (Hank et al., 2015). However, just like in any eco-hydrological model, the agricultural management subroutine simulates the growing seasons through accumulated heat units (a temperature and photoperiodic day-length-dependent approach is used). To use the plant and harvest dates (preferred approach for tropical and subtropical regions), information is required on the cropping calendar and management practices e.g fertilization which highlights the methodological advances presented in this paper. i.e exploitation of the existing global phenological data sets of rainfed and irrigated croplands with the associated cropping calendar and management practices.

- 4a and c show a good improvement of LAI, also seen to some extent in Figs. 5a and c. Also erosion is reduced accordingly. But why do erosion values show strong peaks in March and May in Figs. 5b and d? In March LAI is already high.

**Response:** The strong peaks in March and May can be explained by the reduction of residue on the soil surface. With the second rainy season occurring with no crop cover, part of the residue generated from the first growing season is washed away. Hence with the rainy season occurring in March, we obtain erosion peaks regardless of the relatively high LAI value in the default model that uses a generic uniform crop. However, with the realistic crop representation, we obtain a slightly higher LAI that reduces the erosion peaks in the revised SWAT+ model (we notice a slight increase in the LAI magnitude having a strong impact on the erosion peaks). But with only one growing season, we still have part of the residue washed away in the second rainy season. According to Neitsch et al., (2005), a given residue percentage on the soil surface is more effective than the same percentage of canopy cover. Residue intercepts falling raindrops so near the surface that the drops regain no fall velocity. Additionally, residue obstructs runoff flow, reducing its velocity and transport capacity, [explained in lines 274 – 279]. The explanation will be better phrased in the revised manuscript for clarity.

- Line 191: The idea is clear that results are compared with reality to the extent possible. But what does 'scientific validation' mean? How is this different from just 'validation'?

Biondi et al., (2012) draws a distinction between performance validation and scientific validation. Performance validation is a typical approach adopted to evaluate model performance that requires the comparison between simulated outputs on a set of observations that were not initially used for

calibration. This involves the use of graphical techniques or performance metrics. A scientific validation aims at evaluating the consistency of the model thought as input-state-output system. This concept was derived from the idea that verifying the model performance by simply comparing outputs and observations does not assure that the model is correct from a scientific point of view. This validation may include and extend the performance validation and is specifically required in cases when the quality and quantity of observation data is not sufficient to allow an adequate validation. Studies such as (Haregeweyn et al., 2017) have used the term 'scientific validation' in validating their results. Just like our research, they did the validation by comparing the soil erosion model outputs with previous studies and the scant observations.

However, since the term 'scientific validation' seems to be confusing in this context, it is to be replaced with 'plausibility check' in the revised manuscript.

**Lines 261-263: wouldn't this speak to an increase of ET as water is applied without limits? Please clarify.**

**Response:** No water is applied with limits (50mm every application). The water is irrigated when the water stress is below a specified threshold of 0.7. By unlimited source (which is the deep aquifer in the model), we mean that the water is always available for irrigation when the water stress in the field during the growing season goes below the specific threshold. However in real life scenarios, the farmer may not always have water for irrigation when needed on a specific day due to different management schedules of irrigation schemes. Additionally, not all irrigation water comes from the deep aquifer on all irrigation sites. The statement will be rephrased and a clear explanation added to text in the revised manuscript.

**References**

Biondi, D., Freni, G., Iacobellis, V., Mascaro, G., and Montanari, A.: Validation of hydrological models: Conceptual basis, methodological approaches and a proposal for a code of practice, Physics and Chemistry of the Earth, Parts A/B/C, 42–44, 70–76, https://doi.org/10.1016/j.pce.2011.07.037, 2012.

Burakowski, E., Tawfik, A., Ouimette, A., Lepine, L., Novick, K., Ollinger, S., Zarzycki, C., and Bonan, G.: The role of surface roughness, albedo, and Bowen ratio on ecosystem energy balance in the Eastern United States, Agricultural and Forest Meteorology, 249, 367–376, https://doi.org/10.1016/j.agrformet.2017.11.030, 2018.

Hank, T. B., Bach, H., and Mauser, W.: Using a Remote Sensing-Supported Hydro-Agroecological Model for Field-Scale Simulation of Heterogeneous Crop Growth and Yield: Application for Wheat in Central Europe, 7, 3934–3965, https://doi.org/10.3390/rs70403934, 2015.

Haregeweyn, N., Tsunekawa, A., Poesen, J., Tsubo, M., Meshesha, D. T., Fenta, A. A., Nyssen, J., and Adgo, E.: Comprehensive assessment of soil erosion risk for better land use planning in river basins: Case study of the Upper Blue Nile River, Science of The Total Environment, 574, 95–108, https://doi.org/10.1016/j.scitotenv.2016.09.019, 2017.

Neitsch, S. L., Arnold, J. G., Kiniry, J. R., Williams, J. R., and King, K. W.: SWAT theoretical documentation, 494, 234–235, 2005.

Siad, S. M., Iacobellis, V., Zdruli, P., Gioia, A., Stavi, I., and Hoogenboom, G.: A review of coupled hydrologic and crop growth models, Agricultural Water Management, 224, 105746, https://doi.org/10.1016/j.agwat.2019.105746, 2019.

Williams, J. R. and Singh, V.: The EPIC Model, Computer models of watershed hydrology, 909–1000, 1995.

---

## Author Comment (AC2)

**Reviewer 2**

**General comments**

The manuscript describes how the incorporation of crop specific phenology data improves ET and soil erosion estimates for large-scale simulations using SWAT+ throughout the Nile basin as compared to the default phenology implemented in the model. The simulated LAI and ET values agree much better with validation data obtained from remote sensing. The estimated erosion rates are substantially lower as compared to the default model.

The topic of the manuscript fits the scope of HESS. The results are relevant because they demonstrate the how important it may be to account in an adequate manner for regional differences of crop-specific phenologies.

Unfortunately, the method section is not very well written and is often rather confusing. The key element for the improved model set-up is the use of a global data set on plant and harvest dates for specific crops (Global Gridded Crop Model Intercomparison (GGCMI), see Tab. 1, L. 164 - 168) instead of using the default heat unit approach implemented in SWAT+ by default. It is pointed out in the Abstract and the Introduction that this default approach often fails in tropical regions because crop development is strongly affected by precipitation (e.g., L. 17, 47) while temperature is well suited for temperate regions. However, it remains obscure how the GGCMI data account for this deficiency. It is not explained whether these phenology data are based on observational reference data or on model simulations. If they were model based, one should know how the model accounts for precipitation and temperature compared to the SWAT+ concept. Irrespective of whether the dataset is observation or model based, one should know whether the data represent long-term averages or account for yearly variations. It remains also obscure what the spatial resolution of the dataset is.

These temporal aspects are also neglected in the analysis of the results. The authors used seven years of data for model validation (L. 173). However, they only present results averaged across the entire study period (2009 – 2015). They don't present any data on inter-annual variability (e.g. of precipitation) that might have affected the results. At least in some regions, rainfall varied affecting also the crops simulated in the manuscript (e.g., Epule, Dhiba et al. 2021). Such inter-annual differences can also be expected for erosion, which is very much triggered by few events.

Recommendation

The manuscript requires substantial improvements regarding

1. the presentation of the methods
2. the temporal aspects of the data series that have been analysed.
3. the issues listed in the detailed comments

References:

Epule, T. E., D. Dhiba, D. Etongo, C. Peng and L. Lepage (2021). "Identifying maize yield and precipitation gaps in Uganda." SN Applied Sciences **3**(5): 537.

**Response:** Thank you for a positive evaluation of our study and your time to provide critical comments to improve the manuscript.

The GGCMI crop calendar (Jägermeyr et al., in revision) is an observation-based product, combining first-hand data sources from various agricultural ministries. These data do not include modeling results and therefore overcome the missing temperature-precipitation seasonality in the old model version. In the new model version we schedule planting dates and cultivar selection based on real-world observational planting and harvest data. Planting thus happens at the prescribed day per crop in each 0.5° grid cell. The harvest day varies in each year as physiological maturity depends on accumulated phenological heat units. In a warm growing season maturity is reached earlier than in a cold season. On average, cultivars are selected to match the observational harvest day.

With regards to the temporal aspects of the analysis, we focused on the seasonal representation of the LAI and its subsequent impact on erosion (for illustration of the concept) because as shown in the example Figures 1 and 2 (below), the seasonal pattern is similar in all years. Even though there is some interannual variability in the magnitude of the LAI signal and precipitation, the seasonal pattern is consistent. Consequently, for clarity of the message, we focused on showing the variability in the months (season) within which the LAI and erosion estimates peak since the season is the same throughout the year regardless of the interannual variability in the magnitudes of the signals. Of course, when focusing on the crop outputs such as yield or water productivity, analysis of the interannual variability is very important with the spatial scale of analysis put into consideration. The interannual variability discussion will be highlighted in the revised manuscript.

[Figure]

Figure 1: LAI comparison for rainfed maize in the Blue Nile basin

[Figure]

Figure 2: LAI comparison for rainfed wheat in the Victoria basin

**Detailed comments:**

- L. 87 – 88: Please describe more precisely (in the Method section) what these tables and datasets provide.

**Response:** Decision tables are already described in the 'Method section" from Lines 125 to Line 129. Global datasets of rainfed and irrigated cropland, associated management practices of Fertilization (Nitrogen and Phosphorus) are specified in Table 1, Line 152. However, additional descriptions of the datasets will be added in the revised manuscript for clarity.

- L. 97: Study area: Please describe the study period as well.

**Response:** The study period from 2009 to 2015 is to be further described in the revised manuscript.

- L. 146: "approached suggested by Chawanda": Approach for doing what?

**Response:** Chawanda et al., (2020) proposed an approach for setting up a SWAT+ model using the harmonized land use product (LUH2; Hurtt et al., 2020) ) that was utilized in this study. LUH2 product is formatted as NetCDF with each layer containing a percentage of land use instead of a single-layer raster file. The SWAT+ code has to be adapted to include routines that read the LUH2 NetCDF data as suggested by Chawanda et al., (2020). A brief description of the approach will be added in the revised manuscript.

- L. 152 (Table 1): Confusing: what is used for model set-up and what for comparison between the default and the revised model. Also linguistically. the sentence is strange (data sets are not used for crop management). GGCMI: please provide more details to address the questions mentioned above (general comments).

**Response:** Table 1 only includes datasets used for model setup. Datasets used for model comparison (validation) are described in "section 2.6 Validation of model results – Line 171". A brief description about GCCMI dataset together with other datasets is to be added in the revised manuscript. The caption of Table 1, Line 152 is to be rectified and a separate table showing datasets used in model validation will be added in the revised manuscript.

- L. 153: Section 2.5: It is not clear whether this section presents the revised SWAT version only or the land use for both the default and the revised version. If it's only about the revised version describe how land use was established for the default version. Otherwise, clearly indicate which part only refers to the revised version.

**Response:** The default agricultural land use is precisely stated in the "section 2.4 Default Model set up". By default, the cropland use was represented in a generic way using heat units to trigger the cropping season [Line 148 – line Line149]. This is how SWAT+ represents agricultural land use by default (Arnold et al., 2013). Section 2.5 gives the proposed scheduling using the plant and harvest dates extracted from the global phenology dataset (GCCMI data) and the fertilization extracted from (Hurtt et al., 2020) and (Lu and Tian, 2017) as described in the section. This will be clearly indicated in the revised manuscript.

- L. 164: crop phenology data: please provide more information (see above).

**Response:** More information about the global phenology dataset (GGCMI data) is to be added in the revised manuscript.

- L. 191: What is a scientific validation? As it reads later in the manuscript it seems to be a plausibility check.

Response: Biondi et al., (2012) draws a distinction between performance validation and scientific validation. Performance validation is a typical approach adopted to evaluate model performance that requires the comparison between simulated outputs on a set of observations that were not initially used for calibration. This involves the use of graphical techniques or performance metrics. A scientific validation aims at evaluating the consistency of the model thought as input-state-output system. This concept was derived from the idea that verifying the model performance by simply comparing outputs and observations does not assure that the model is correct from a scientific point of view. This validation may include and extend the performance validation and is specifically required in cases when the quality and quantity of observation data is not sufficient to allow an adequate validation. Studies such as (Haregeweyn et al., 2017) have used the term 'scientific validation' in validating their results. Just like our research, they did the validation by comparing the soil erosion model outputs with previous studies and the scant observations.

However, since the term 'scientific validation' seems to be confusing in this context, it is to be replaced with 'plausibility check' in the revised manuscript.

- L. 194: What's the meaning of "moreover" at that point?

Response: The intended meaning was "in addition to the previous statement". However, "moreover" is to be omitted and the statement rephrased in the revised manuscript.

- L. 197 – 200: These two sentences are not clear.

Response: The statements are to be rephrased but what they intend to mean is that in order to isolate the uncertainty in the default and revised model setups due to agricultural land use representation, the models were compared in default parameters considering that calibrations change with different catchments. So only the water balance was checked to ensure that we have consistency of the model thought as an input-state-output system. This means that the differences seen in the model setups originate primarily from the crop representation and management practices. However, the two statements will be rephrased for clarity in the revised manuscript.

L. 204 – 207: How has the model be parameterized? No information is provided. How well did the model perform e.g., when compared to discharge? The water balance information provided are not very conclusive since L. 254 demonstrates that the model can be quite off for one component (ET) while still closing the water balance well (compensating errors).

Response: A default parameterization of the model was used, (this will be added in the revised manuscript). The study aims at improving the default model simulation by better representing the physical land processes (crop growth and evapotranspiration), prior to any calibration. A flow calibration would not much affect these processes but rather aims at improving flow simulations, which was not the objective of our study. We only focused on the ET component of the water balance to show how representation of crop phenology can impact the ET estimates which should ideally precede any calibration efforts [Line 200 – Line 203]. Besides, SWAT was developed with the objective of predicting the impact of management on water, sediment and agricultural yields in large 'ungauged' basins where usually no data is available for calibrations (Arnold et al., 1998; Srinivasan et al., 2010). Additionally, matching flow observations at the outlet/gauging station does not necessarily mean that the internal processes e.g ET are realistically simulated.

- L. 208: Results: Please provide a short description of the hydro-climatic characterization of the study period (2009 – 2015) including metrics of temporal variability.

**Response:** The hydro-climatic description of the study period is to be added in the "Results" section of the revised manuscript.

- L. 210 – 220: Provide information about variability and model performance across years as well.

**Response:** We originally focused on the seasonal variations of the LAI and erosion estimates as we believe the annual cycle information is relevant for highlighting the value of this representation. For example other studies such as Levis et al., (2012) that incorporated agriculture in a Community Earth System Model (CESM1), evaluated vegetation traits e.g. LAI, on a seasonal/annual basis. However, the interannual variability will also be discussed in the revised manuscript.

- L. 224: Fig. 4: Indicate in the caption whether the data represent average values. If they are provide standard deviations in the figures.

**Response:** The standard deviation is to be added in the revised manuscript for data representing averages and captions will be adjusted.

- L. 226: Fig. 5: Same comment as above. Additionally: the LAI of both models are rather similar. How does it come that the erosion rates differed so strongly? Explain.

**Response:** We noticed that a slight increase in the LAI magnitude had a strong impact on the erosion peaks. Even though the cropping season in both the default and revised model setups captures only one cropping season, there is still a reduction in the HRU erosion estimates because the revised SWAT+ LAI, representative of an actual crop is slightly greater than the default LAI representative of a generic crop. Additionally, with a slightly higher LAI magnitude in the revised SWAT+ model, more biomass is generated which results in more residue that could be more effective in reducing soil erosion even after the cropping season. Residue intercepts rain droplets near the soil surface that drops regain no fall velocity. Thus, a given percentage of residue is more effective than the same percentage of canopy cover (Neitsch et al., 2011).(lines 274-279). This will be explicitly explained in the revised manuscript.

L. 230: The situations are probably denoted by Cases 1 and 2 in Fig. A2. However, this is not explained. How could one differentiate between the two cases in the spatial data?

**Response:** Cases 1 refer to growing season (irrigated) occurring in the dry season of the season while case 2 refers to growing season (irrigated) occurring during the first rainy season. So case 2 is mainly supplementary irrigation. This information is to be added to text in the revised manuscript.

- L. 240: Same comment as for Fig. 4; Additionally: Why is LAI for the default so low? Was there an underestimation of irrigated wheat (acreage)? If yes, this would imply that not only phenology but also land use differed between model versions. This was not made explicit so far. Clarify.

**Response:** By default, no management practices (i.e. irrigation and fertilization) are implemented. Hence, being in the Nile delta that predominantly relies on irrigation for plant growth, the plant growth is constrained by default causing the low LAI. However, with the implemented crop phenology and associated management practices, we see an improvement in the LAI simulation. This information will be added to text in the revised manuscript.

- L. 260 – 263: Would one not expect an overestimation of irrigation and therefore ET from an ideal unlimited water source for irrigation?

**Response:** Water is applied with limits (50mm every application for this case study) which avoids the continuous over irrigation. The water is irrigated when the water stress is below a specified threshold of 0.7. By unlimited source (which is the deep aquifer in the model), we mean that the water is always available for irrigation when the water stress in the field during the growing season goes below the specific threshold. However in real life scenarios, the farmer may not always have water for irrigation when needed on a specific day due to different management schedules of irrigation schemes. . Additionally, not all irrigation water comes from the deep aquifer on all irrigation sites. The statement will be rephrased and explained better in the revised manuscript.

- L. 266: Fig. 7: A difference map between model predictions and remote sensing observations would be more instructive. How can one distinguish between agricultural and non-agricultural ET? Units: correct to mm $y^{-1}$.

**Response:** Fig.7 in the manuscript masks out non-agricultural areas and presents only ET for agricultural areas to highlight the impact of the agricultural areas since this study focused on the ET response to agricultural land use representation. Presenting agricultural ET further highlights the impact of human activities (planting, harvesting, irrigation, fertilization) which is significant for developing agricultural water resources management strategies (Wang et al., 2008). However, a difference map on model ET output will be included too in the revised manuscript. The units will be corrected as well in the revised manuscript.

- L280 – 283: Below, a specific comparison is presented for the Blue Nile region. Please provide also the relative change for this area to help the reader linking the two aspects.

**Response:** The relative change in the Upper Blue Nile is to be provided in the revised manuscript.

- L. 297: On L. 295 a max of 20500 t $km^{-2}$ $y^{-1}$ is mentioned. Can you explain?

**Response:** The simulated annual average soil erosion ranged from 0 to 20500 t $km^{-2}$ $y^{-1}$ in the whole region. The standard deviation is to be added for this value in the revised manuscript. On Line 297, we mention an annual average soil erosion range of 0 to 13000 t $km^{-2}$ $y^{-1}$ in the Upper Blue Nile basin. The statements will be rephrased in the revised manuscript.

**References**

Arnold, J. G., Srinivasan, R., Muttiah, R. S., and Williams, J. R.: Large Area Hydrologic Modeling and Assessment Part I: Model Development1, 34, 73–89, https://doi.org/doi:10.1111/j.1752-1688.1998.tb05961.x, 1998.

Arnold, J. G., Kiniry, J. R., Srinivasan, R., Williams, J. R., Haney, E. B., and Neitsch, S. L.: SWAT 2012 input/output documentation, Texas Water Resources Institute, 2013.

Biondi, D., Freni, G., Iacobellis, V., Mascaro, G., and Montanari, A.: Validation of hydrological models: Conceptual basis, methodological approaches and a proposal for a code of practice, Physics and Chemistry of the Earth, Parts A/B/C, 42–44, 70–76, https://doi.org/10.1016/j.pce.2011.07.037, 2012.

Chawanda, C. J., Arnold, J., Thiery, W., and Griensven, A. van: Mass balance calibration and reservoir representations for large-scale hydrological impact studies using SWAT+, Climatic Change, 1–21, https://doi.org/10.1007/s10584-020-02924-x, 2020.

Haregeweyn, N., Tsunekawa, A., Poesen, J., Tsubo, M., Meshesha, D. T., Fenta, A. A., Nyssen, J., and Adgo, E.: Comprehensive assessment of soil erosion risk for better land use planning in river basins: Case study of the Upper Blue Nile River, Science of The Total Environment, 574, 95–108, https://doi.org/10.1016/j.scitotenv.2016.09.019, 2017.

Hurtt, G. C., Chini, L., Sahajpal, R., Frolking, S., Bodirsky, B. L., Calvin, K., Doelman, J. C., Fisk, J., Fujimori, S., Goldewijk, K. K., Hasegawa, T., Havlik, P., Heinimann, A., Humpenöder, F., Jungclaus, J., Kaplan, J., Kennedy, J., Kristzin, T., Lawrence, D., Lawrence, P., Ma, L., Mertz, O., Pongratz, J., Popp, A., Poulter, B., Riahi, K., Shevliakova, E., Stehfest, E., Thornton, P., Tubiello, F. N., van Vuuren, D. P., and Zhang, X.: Harmonization of Global Land-Use Change and Management for the Period 850–2100 (LUH2) for CMIP6, 1–65, https://doi.org/10.5194/gmd-2019-360, 2020.

Jägermeyr et al.: Climate change signal in global agriculture emerges earlier in new generation of climate and crop models, in revision.

Levis, S., Bonan, G. B., Kluzek, E., Thornton, P. E., Jones, A., Sacks, W. J., and Kucharik, C. J.: Interactive Crop Management in the Community Earth System Model (CESM1): Seasonal Influences on Land–Atmosphere Fluxes, 25, 4839–4859, https://doi.org/10.1175/JCLI-D-11-00446.1, 2012.

Lu, C. C. and Tian, H.: Global nitrogen and phosphorus fertilizer use for agriculture production in the past half century: shifted hot spots and nutrient imbalance, 9, 181, 2017.

Neitsch, S. L., Arnold, J. G., Kiniry, J. R., and Williams, J. R.: Soil and water assessment tool theoretical documentation version 2009, Texas Water Resources Institute, 2011.

Srinivasan, R., Zhang, X., and Arnold, J.: SWAT ungauged: hydrological budget and crop yield predictions in the Upper Mississippi River Basin, 53, 1533–1546, 2010.

Wang, S., Kang, S., Zhang, L., and Li, F.: Modelling hydrological response to different land-use and climate change scenarios in the Zamu River basin of northwest China, 22, 2502–2510, 2008.

---

## Author Response (AR1)

**Point-by-point response to the Editor (Dr. Christian Stamm) and the two anonymous referee comments and suggestions**

**General authors note:** Thank you for a positive evaluation of our study and we very much appreciate all the comments from the editor and the two anonymous reviewers. We believe the comments and suggestions have helped us to improve the clarity and structure of the revised manuscript.

**Important changes:** The first paragraph of the 'Introduction' section has been re-edited and improved. The 'methodology' section has been improved and restructured with a clear description of the datasets especially the Global Gridded Crop Model Intercomparison (GGCMI) dataset. The 'results and discussion' section has also been improved with the inclusion of a brief description of the spatial and temporal analysis of the rainfall (sub-section: 3.1), the temporal analysis of the LAI and erosion (Figure 9). Additionally, all figures have been improved and three new figures (Figure 4, Figure 5 and Figure 9) have been added. Overall, we have further improved the language throughout the revised manuscript.

Below we provide a point-by-point response in blue (italics).
* * *
**Response to the Editor**

**Comments to the Author:**

Thank you for responding to the comments provided by the two reviews on your manuscript "Improved Representation of Agricultural Land Use and Crop Management for Large Scale Hydrological Impact Simulation in Africa using SWAT+". The reviews indicate that major revisions are necessary. Please revise the manuscript according to your suggestions and pay attention to properly address all issues. Please provide a point-by-point explanation of your revisions.

*Response: We thank the editor for the assessment and comments for our manuscript. We have revised the manuscript according to the suggestions and addressed all the issues as required. The point-by-point responses are provided below in italics and the changes in the revised manuscript (added below) are in blue.*
* * *
**Response to Reviewer 1**

The paper presents an important step forward to better capture crop phenology in hydrological modelling. It is well written and structured. To demonstrate its reach and inspiration for future modelling, the paper could be better positioned with regard to existing modelling advances considering crop dynamics. It would gain from a discussion of results in view of existing modelling approaches and advances, going beyond the mention of omitting these in most models. Discussing uncertainties arising from the sole focus of the most widespread crop per pixel and simulation of the 5 representative crops would also make the paper stronger.

*Response: Thank you for a positive evaluation of our study and your time to provide critical comments to improve the manuscript.*

*A discussion of results (LAI and ET) in view of existing approaches has been added in the revised manuscript. Specifically, the LAI results have been discussed in reflection of other approaches that have improved phenology development by using remote sensing LAI datasets (Ma et al., 2019; Rajib et al., 2020). Although the use of remote sensing improves the LAI simulations, the challenge of local knowledge is still required for crop type and crop management mapping (Bégué et al., 2018). Additionally, the potential of improving phenology datasets to include multiple cropping seasons has been discussed in regards to advancements in research regarding cropping patterns such as the crop generator (Sietz et al., 2021) and mapped global areas of different cropping systems (Waha et al., 2020), [Lines 320 – 340].*

*As regards to the ET simulations, the improvement in ET estimates has been discussed in relation to incorporating temporal patterns of LAI that favor transpiration and evaporation from canopy intercepted water [Lines 347 - 351]. Uncertainties arising from sole focus on the widespread crop per pixel as compared to the simulation of the 5 representative crops has been added under the discussion of ET simulations. This is because the heterogenous representation of crops in a pixel can have effects on the ET fluxes due to variations in the physical characteristics of vegetation such as root depth, LAI, stomata conductance, surface roughness and albedo (Burakowski et al., 2018) [Lines 363 - 367].*

Further remarks

- The start of the introduction would be more interesting to read if it wasn't a copy of the abstract.

*Response: Yes, the start of the introduction has been modified accordingly, [Lines 33 - 47].*

- Lines 34-35: some details of the "simplistic way"/"abstract terms" of considering crops in hydrological models and what we learn from these would provide a good start.

*Response: With the modification of the start of introduction, 'simplistic/abstract terms' have been omitted and rather 'a uniform generic crop' adopted that is usually used in most regional hydrologic models( Schuol and Abbaspour, 2006; Schuol et al., 2008; Chawanda et al., 2020). Further details are provided in the introduction section [Lines 33 - 47].*

- Line 62: examples on current knowledge derived from the implementation of cropping dynamics in hydrological modelling would be useful (e.g. The Crop Generator: Implementing crop rotations to effectively advance eco-hydrological modelling. https://www.sciencedirect.com/science/article/abs/pii/S0308521X21001360).

*Response: Yes, examples on current knowledge about approaches on representation of crop dynamics in other eco-hydrological models (e.g Sietz et al., 2021; Zhang et al., 2021) has been added, [Lines 63 - 67].*

- Lines 70-75: How is phenological crop development simulated in other hydrological models, e.g. SWIM? Has there been any coupling attempts with modelling of land surface processes that consider phenology, e.g. PROMET?

*Response: Several hydrological models e.g SWAT+, SWIM have an integration of a hydrologic module and crop growth module (mostly EPIC, Williams and Singh, 1995) and by default, crop growth is simulated by accumulated heat units. In most applications, modelers use a generic or a single crop for representation of agricultural land use. As highlighted, the challenge of this representation is twofold; in tropical and subtropical regions, the use of heat units easily causes incorrect cropping seasons as these regions are primarily driven by water availability. Secondly, the use of a generic crop fails to account for any variability in existing heterogenous agricultural land use. A review of coupled hydrologic and crop growth models has been done by Siad et al., (2019).*

*Coupling attempts of land surface process models e.g PROMET to consider phenology have been done through an agricultural management subroutine which also initializes management actions that may be of hydrologic, climatic or agricultural consequence (Hank et al., 2015). However, just like in any eco-hydrological model, the agricultural management subroutine simulates the growing seasons through accumulated heat units (a temperature and photoperiodic day-length-dependent approach is used). To use the plant and harvest dates (preferred approach for tropical and subtropical regions), information is required on the cropping calendar and management practices e.g fertilization which highlights the methodological advances presented in this paper. i.e exploitation of the existing global phenological data sets of rainfed and irrigated croplands with the associated cropping calendar and management practices.*

- 4a and c show a good improvement of LAI, also seen to some extent in Figs. 5a and c. Also erosion is reduced accordingly. But why do erosion values show strong peaks in March and May in Figs. 5b and d? In March LAI is already high.

*Response: The strong peaks in March and May can be explained by the reduction of residue on the soil surface. With the second rainy season occurring with no crop cover, part of the residue generated from the first growing season is washed away. Hence with the rainy season occurring in March, we obtain erosion peaks regardless of the relatively high LAI value in the default model that uses a generic uniform crop. However, with the realistic crop representation, we obtain a slightly higher LAI that reduces the erosion peaks in the revised SWAT+ model (we notice a slight increase in the LAI magnitude having a strong impact on the erosion peaks). But with only one growing season, we still have part of the residue washed away in the second rainy season. According to Neitsch et al., (2005), a given residue percentage on the soil surface is more effective than the same percentage of canopy cover. Residue intercepts falling raindrops so near the surface that the drops regain no fall velocity. Additionally, residue obstructs runoff flow, reducing its velocity and transport capacity, [Lines 379 – 388].*

- Line 191: The idea is clear that results are compared with reality to the extent possible. But what does 'scientific validation' mean? How is this different from just 'validation'?

*Response: Biondi et al., (2012) draws a distinction between performance validation and scientific validation. Performance validation is a typical approach adopted to evaluate model performance that requires the comparison between simulated outputs on a set of observations that were not initially used for calibration. This involves the use of graphical techniques or performance metrics. A scientific validation aims at evaluating the consistency of the model thought as input-state-output system. This concept was derived from the idea that verifying the model performance by simply comparing outputs and observations does not assure that the model is correct from a scientific point of view. This validation may include and extend the performance validation and is specifically required in cases when the quality and quantity of observation data is not sufficient to allow an adequate validation. Studies such as (Haregeweyn et al., 2017) have used the term 'scientific validation' in validating their results. Just like our research, they did the validation by comparing the soil erosion model outputs with previous studies and the scant observations.*

*However, since the term 'scientific validation' seems to be confusing in this context, it has been replaced with 'plausibility check' in the revised manuscript, [Lines 236 – 237].*

- Lines 261-263: wouldn't this speak to an increase of ET as water is applied without limits? Please clarify.

***Response**: Thank you for pointing this out. However, water is applied with limits (50mm every application). The water is irrigated when the water stress is below a specified threshold of 0.7. By unlimited source (which is the deep aquifer in the model), we mean that the water is always available for irrigation when the water stress in the field during the growing season goes below the specific threshold. However in real life scenarios, the farmer may not always have water for irrigation when needed on a specific day due to different management schedules of irrigation schemes. Additionally, not all irrigation water comes from the deep aquifer on all irrigation sites. The statement has been rephrased and a clear explanation added to text in the revised manuscript, [Lines 356 - 359].*

**Response to Reviewer 2**

**General comments**

The manuscript describes how the incorporation of crop specific phenology data improves ET and soil erosion estimates for large-scale simulations using SWAT+ throughout the Nile basin as compared to the default phenology implemented in the model. The simulated LAI and ET values agree much better with validation data obtained from remote sensing. The estimated erosion rates are substantially lower as compared to the default model.

The topic of the manuscript fits the scope of HESS. The results are relevant because they demonstrate the how important it may be to account in an adequate manner for regional differences of crop-specific phenologies.

Unfortunately, the method section is not very well written and is often rather confusing. The key element for the improved model set-up is the use of a global data set on plant and harvest dates for specific crops (Global Gridded Crop Model Intercomparison (GGCMI), see Tab. 1, L. 164 - 168) instead of using the default heat unit approach implemented in SWAT+ by default. It is pointed out in the Abstract and the Introduction that this default approach often fails in tropical regions because crop development is strongly affected by precipitation (e.g., L. 17, 47) while temperature is well suited for temperate regions. However, it remains obscure how the GGCMI data account for this deficiency. It is not explained whether these phenology data are based on observational reference data or on model simulations. If they were model based, one should know how the model accounts for precipitation and temperature compared to the SWAT+ concept. Irrespective of whether the dataset is observation or model based, one should know whether the data represent long-term averages or account for yearly variations. It remains also obscure what the spatial resolution of the dataset is.

These temporal aspects are also neglected in the analysis of the results. The authors used seven years of data for model validation (L. 173). However, they only present results averaged across the entire study period (2009 – 2015). They don't present any data on inter-annual variability (e.g. of precipitation) that might have affected the results. At least in some regions, rainfall varied affecting also the crops simulated in the manuscript (e.g., Epule, Dhiba et al. 2021). Such inter-annual differences can also be expected for erosion, which is very much triggered by few events.

Recommendation

The manuscript requires substantial improvements regarding

1. the presentation of the methods
2. the temporal aspects of the data series that have been analysed.
3. the issues listed in the detailed comments

*Response: Thank you for pointing this out. However, water is applied with limits (50mm every application). The water is irrigated when the water stress is below a specified threshold of 0.7. By unlimited source (which is the deep aquifer in the model), we mean that the water is always available for irrigation when the water stress in the field during the growing season goes below the specific threshold. However in real life scenarios, the farmer may not always have water for irrigation when needed on a specific day due to different management schedules of irrigation schemes. Additionally, not all irrigation water comes from the deep aquifer on all irrigation sites. The statement has been rephrased and a clear explanation added to text in the revised manuscript, [Lines 356 - 359].*

- L. 266: Fig. 7: A difference map between model predictions and remote sensing observations would be more instructive. How can one distinguish between agricultural and non-agricultural ET? Units: correct to mm y$^{-1}$.

*Response: Updated to Fig.10 in the manuscript masks out non-agricultural areas and presents only ET for agricultural areas to highlight the impact of the agricultural areas since this study focused on the ET response to agricultural land use representation. Presenting agricultural ET further highlights the impact of human activities (planting, harvesting, irrigation, fertilization) which is significant for developing agricultural water resources management strategies (Wang et al., 2008). However, a difference map (revised model ET – WaPOR ET) has also been added to highlight the regions that still have ET underestimations (mostly areas that are irrigated with multiple cropping practices). The units have been corrected in the revised manuscript.*

- L280 – 283: Below, a specific comparison is presented for the Blue Nile region. Please provide also the relative change for this area to help the reader linking the two aspects.

*Response: The relative change in the Upper Blue Nile has been added to the revised manuscript, [Lines 396 - 397].*

- L. 297: On L. 295 a max of 20500 t km$^{-2}$ y$^{-1}$ is mentioned. Can you explain?

*Response: The simulated annual average soil erosion ranged from 0 to over 20500 t km$^{-2}$ y$^{-1}$ in the whole region revealing the severity of soil erosion in the Ethiopian highlands (Fig. 12) as compared to the other parts of the region. Ethiopian highlands have been reported to have high soil erosion and sediment yield rates attributed to partly topography and rainfall but also to recent and historic land conversions to agriculture. The text has been revised and improved in the revised manuscript, [Lines 414 - 417].*
* * *
References

[revised manuscript text omitted]

Wang, S., Kang, S., Zhang, L., and Li, F.: Modelling hydrological response to different land-use and climate change scenarios in the Zamu River basin of northwest China, 22, 2502–2510, 2008.

[revised manuscript text omitted]

---

## Author Response (AR2)

**Reply to Editor**

Dear Dr. Nkwasa,

Thank you for the revision of the manuscript. The reviewers were very positive with the new version. There were only some minor aspects that one reviewer brought up (see below). Please respond to these comments.

Sincerely,

Christian Stamm, Editor HESS

General authors' note: Thank you for a positive evaluation of the revised manuscript. We appreciate all the feedback and we are glad that the clarity of the manuscript has greatly improved.

Comments Reviewer 1:

The authors have done a very serious and good job in addressing the issues raised in the review. The manuscript reads much better now. The methods and approach are well described and one can easily grasp the rationale. The authors have also added additional data and results and expanded on the discussion where necessary. This substantially helps to fully understand the findings.

I still have some comments but they are mostly suggestions for issues that might be (briefly) discussed in the paper:

Specific comments:

-Fig. 1: What is the data source for the map? Shouldn't that be indicated?

Response: Thank you for your positive evaluation of our revised manuscript. As regards to Fig.1, the map is as a result of the delineation done in this study. The source of the DEM used in the delineation has already been provided in the text in Table 1.

-Fig-7: if I interpret correctly, the actual LAI in the Victoria basin is most of the time much higher than the calculated LAI. One explanation could be the small-scale farming in the region where farmers cultivate on very small crops and often have intercropping of many different crops. Therefore, the spatial assignment of one single crop (with its easonal growth patterns driving the LAI in the model) may be only a poor representation of reality. So in addition to lack of a second crop that is mentioned in the manuscript (a limitation of the existing data in a temporal sense), there would also be a limitation in a spatial sense. Please comment on that aspect. It indeed supports your conclusion that local knowledge is required to grasp the complexity of the cropping patterns.

Response: Yes indeed, by using a single crop to represent a pixel, we have a limitation in a spatial sense. Especially, in the Victoria basin that has several combinations of intercropping and sequential cropping at plot scale. This has been highlighted in the text as a limitation that needs addressing especially at local scale application by considering the heterogeneity of cropping patterns in a pixel. [Lines 367 – 371]

I assume that LAI underestimation by the model has also consequences for the ET estimates. Could the widespread presence of "minor" crops be one of the reasons for the underestimation of ET in the Victoria basin? Can you comment on that?

Response: Yes, this has been mentioned as a possible cause of ET underestimation. However, instead of using "minor crops", we have used "crop heterogeneity" in a pixel. The simplification of using a single crop per pixel could have effects on the ET fluxes due to the simplifications in the variations of the physical characteristics (e.g LAI, root depth, stomata conductance) of the heterogenous crops (Burakowski et al., 2018). Additionally, this

simplification alters the partitioning of sensible heat fluxes to latent heat fluxes (Eltahir, 1998) that in turn affect the ET estimates [Lines 367 – 371]

-General comment: By relying on observed data for crop phenology, I'd assume that limits the application of the approach to the past, hence periods when such observations exist. Can you comment on how to deal with forecasting or using such models for prospective climate change studies? Do you see potential get better phenology data also for such purposes?

Response: We are glad that you raised this comment. We believe there is potential to get better phenology data for future impact studies. As highlighted in the manuscript, Waha et al. (2020) mapped global areas of different multiple cropping systems. This can potentially be combined with global phenology datasets to generate crop calendars with different cropping systems. Additionally, as suggested by Kim et al. (2021), future research should be directed towards hybridization of multiple sources of information (e.g satellite data products, model-based products and census-based data), improvements to temporal coverage and resolution, enrichment of management variables and exploration of new sources of information to achieve better phenology datasets.

We have shifted text from line 336 – 338 to the conclusion and recommendation section (Line 454 – 456) as a recommendation for future research to get better phenology data.

Note: We have updated the citation (Jägermeyr et al., in revision) to (Jägermeyr et al., 2021) and updated the reference list.

**References**

Burakowski, E., Tawfik, A., Ouimette, A., Lepine, L., Novick, K., Ollinger, S., Zarzycki, C., Bonan, G., 2018. The role of surface roughness, albedo, and Bowen ratio on ecosystem energy balance in the Eastern United States. Agric. For. Meteorol. 249, 367–376. https://doi.org/10.1016/j.agrformet.2017.11.030

Eltahir, E.A.B., 1998. A Soil Moisture–Rainfall Feedback Mechanism: 1. Theory and observations. Water Resour. Res. 34, 765–776. https://doi.org/10.1029/97WR03499

Jägermeyr et al., in revision. Climate change signal in global agriculture emerges earlier in new generation of climate and crop models. Nat. Food.

Jägermeyr, J., Müller, C., Ruane, A.C., Elliott, J., Balkovic, J., Castillo, O., Faye, B., Foster, I., Folberth, C., Franke, J.A., Fuchs, K., Guarin, J.R., Heinke, J., Hoogenboom, G., Iizumi, T., Jain, A.K., Kelly, D., Khabarov, N., Lange, S., Lin, T.-S., Liu, W., Mialyk, O., Minoli, S., Moyer, E.J., Okada, M., Phillips, M., Porter, C., Rabin, S.S., Scheer, C., Schneider, J.M., Schyns, J.F., Skalsky, R., Smerald, A., Stella, T., Stephens, H., Webber, H., Zabel, F., Rosenzweig, C., 2021. Climate impacts on global agriculture emerge earlier in new generation of climate and crop models. Nat. Food 2, 873–885. https://doi.org/10.1038/s43016-021-00400-y

Kim, K.-H., Doi, Y., Ramankutty, N., Iizumi, T., 2021. A review of global gridded cropping system data products. Environ. Res. Lett. 16, 093005. https://doi.org/10.1088/1748-9326/ac20f4

Waha, K., Dietrich, J.P., Portmann, F.T., Siebert, S., Thornton, P.K., Bondeau, A., Herrero, M., 2020. Multiple cropping systems of the world and the potential for increasing cropping intensity. Glob. Environ. Change 64, 102131. https://doi.org/10.1016/j.gloenvcha.2020.102131